# Vegetated Target Decorrelation in SAR and Interferometry: Models, Simulation, and Performance Evaluation

**Andrea Monti-Guarnieri [1],\* [iD], Marco Manzoni [1] [iD], Davide Giudici [2] [iD], Andrea Recchia [2] and Stefano Tebaldini [1] [iD]**

[1]  Dipartimento di Elettronica, Informazione e Bioingegneria, Politecnico di Milano, Piazza Leonardo da Vinci 32, 20133 Milan, Italy; marco.manzoni@polimi.it (M.M.); stefano.tebaldini@polimi.it (S.T.)

[2]  Aresys srl, via Flumendosa 16, 20132 Milan, Italy; davide.giudici@aresys.it (D.G.); andrea.recchia@aresys.it (A.R.)

\*  Correspondence: andrea.montiguarnieri@polimi.it; Tel.: +39-02-23993446

**Abstract:** The paper addresses the temporal stability of distributed targets, particularly referring to vegetation, to evaluate the degradation affecting synthetic aperture radar (SAR) imaging and repeat-pass interferometry, and provide efficient SAR simulation schemes for generating big dataset from wide areas. The models that are mostly adopted in literature are critically reviewed, and aim to study decorrelation in a range of time (from hours to days), of interest for long-term SAR, such as ground-based or geosynchronous, or repeat-pass SAR interferometry. It is shown that none of them explicitly account for a decorrelation occurring in the short-term. An explanation is provided, and a novel temporal decorrelation model is proposed to account for that fast decorrelation. A formal method is developed to evaluate the performance of SAR focusing, and interferometry on a homogenous, stationary scene, in terms of Signal-to-Clutter Ratio (SCR), and interferometric coherence. Finally, an efficient implementation of an SAR simulator capable of handling the realistic case of heterogeneous decorrelation over a wide area is discussed. Examples are given by assuming two geostationary SAR missions in C and X band.

**Keywords:** SAR focusing; SAR interferometry; decorrelation; geosynchronous SAR; Ground-Based Radar

## 1. Introduction

The generation of both repeat-pass radar interferograms and focused synthetic aperture images requires the stability of target amplitudes and phase, with time. This implies that the target does not move, even by a small fraction of the wavelength (usually a few millimeters). When the echo at one range bin is the contribution of many scatterers in the resolution cell—that is the case for a homogenous target—a temporal variation of the position of each target is enough to cause either defocusing (in the case of synthetic aperture) or decorrelation (in the case of interferometry).

The resulting clutter noise or clutter decorrelation has been widely studied in literature for over a century, initially focusing on fast-moving clutter, such as sea-ocean [1] or windblown forests [2]; thereafter, addressing the impact of moving targets in interferometry [3], in the long-term, up to many days [4]. The clutter models so far derived have been used to evaluate performance, in systems such as airborne and spaceborne radar and synthetic aperture radar (SAR), to develop clutter-suppression methods, and to provide simulators and testbeds for validation and performance optimization.

The assessment of radar decorrelation for vegetated targets is well established in literature in two cases—of very short-term (due to wind) or very long-term. However, there is presently an interest in decorrelation over the medium term, from seconds to hours, affecting Ground-Based Radar (GBR)

or geosynchronous (GEO) Interferometric SAR [5,6]. In such systems, the unpredictable, random variation of the target phase center during the integration time causes impairments in the focusing quality. Such variations are due to several effects, from the propagation in the atmosphere to the changes in vegetation due to the wind, moisture sap flow, etc. The atmospheric contribution has been addressed for both GBR and GEO SAR [7–9], as that will address the scene stability.

The purpose of this paper is to analyze and propose models addressing decorrelation, primarily on vegetated areas, in intervals from hours to days, to propose a suitable model, and to show how this model can be used for both performance analysis and SAR simulation.

More specifically, in the second section, we analyze the most widely adopted models in radar and interferometry, we review these models basing on literature and stability analysis made by GBR and satellite SAR data, and we propose one, namely Sum of Exponentials (SoE), which generalizes the exponential decorrelation used in long-term SAR interferometry [10–15]. We show how the model parameters can be transformed to fit the other models.

In the third section, we derive a performance model to evaluate the impact of speckle decorrelation, both on the SAR images and on repeat pass interferograms, under the assumption of homogenous clutter, then extending the one in [9]. The performance will depend on the Doppler spectrum of the scene and, for the proposed SoE model, a closed-form expression is achieved. In particular, we show that as scene decorrelation affects both focusing and interferometry, the two effects are connected, and the interferometric coherence will not depend upon the clutter noise in the focused image.

In the last section, we show how the generation of decorrelated data can be efficiently embedded for simulating a SAR acquisition with long integration time and interferometry. The time domain-based simulator is capable to cope with the realistic scenario of heterogeneous targets. Examples of simulation and focusing of wide-area scenes, observed by geosynchronous SAR in C and X band are provided.

## 2. Models for Speckle Decorrelation

Three models will be considered here for distributed target decorrelation over a homogenous target. They were developed for different applications and different time scales.

### 2.1. The ICM Model

The Intrinsic Clutter Model (ICM) was first proposed by Billingsley and Larrabee in [16] for windblown Doppler Power Spectrum Density (PSD) of radar clutter observed from low grazing angles. The PSD is composed by two terms: a Dirac pulse, representing the stable contributions, loosely affected by wind, and a low-pass exponential decay, whose slope depends on the wind speed and the radar wavelength, $\lambda$:

$$S_{ICM}(f_D) = \frac{\alpha}{\alpha+1}\delta(f_D) + \frac{1}{\alpha+1}\frac{\lambda\beta}{4}\exp\left(-\lambda\beta\frac{|f_D|}{2}\right),$$ (1)

where $f_D$ is the Doppler and $\alpha$ and $\beta$ are the model parameters that are related to the wind speed, w, and the radar frequency, $f_c$, by the empirical laws:

$$\begin{aligned}\alpha &= 489.9\cdot(w*2.2369)^{-1.55}f_c^{-1.21},\\ \beta &= \frac{1}{(0.1048*(\log_{10}(w*2.2369)+0.4147))},\end{aligned}$$ (2)

$w$ being expressed in meter per second and $f_c$ in GHz.

The ICM model (1) is so conceived that the total power is unitary:

$$\int S_{ICM}(f)\,df = \frac{\alpha}{\alpha+1} + \frac{1}{\alpha+1}\frac{\lambda\beta}{4}\int\exp\left(-\lambda\beta\frac{|f|}{2}\right)df = P_{DC} + P_{AC} = 1,$$ (3)

where $P_{DC}$ and $P_{AC}$ represent the power of the stable part and the clutter. This means that $\alpha$ is the - ratio between the stable power and the clutter, whereby $\beta$ accounts for the spectral slope.

The model has been found quite good over a very wide range of frequency bands from VHF to K, and for times up to hundreds of milliseconds [2,16,17]. Its validity has been widely acknowledged also for targets different from a forest. In Figure 1, a sketch of the ICM model is represented together with two Doppler measures achieved in a Ku-band GBR campaign over a rural landscape. There, the Doppler spectra of two targets at different ranges are chosen as representative of a stable and a decorrelating one. The PSD resembles quite well the exponential shape (1) with the Dirac pulse for stable contribution, but a constant noise floor is to be added to account for thermal and other white noises.

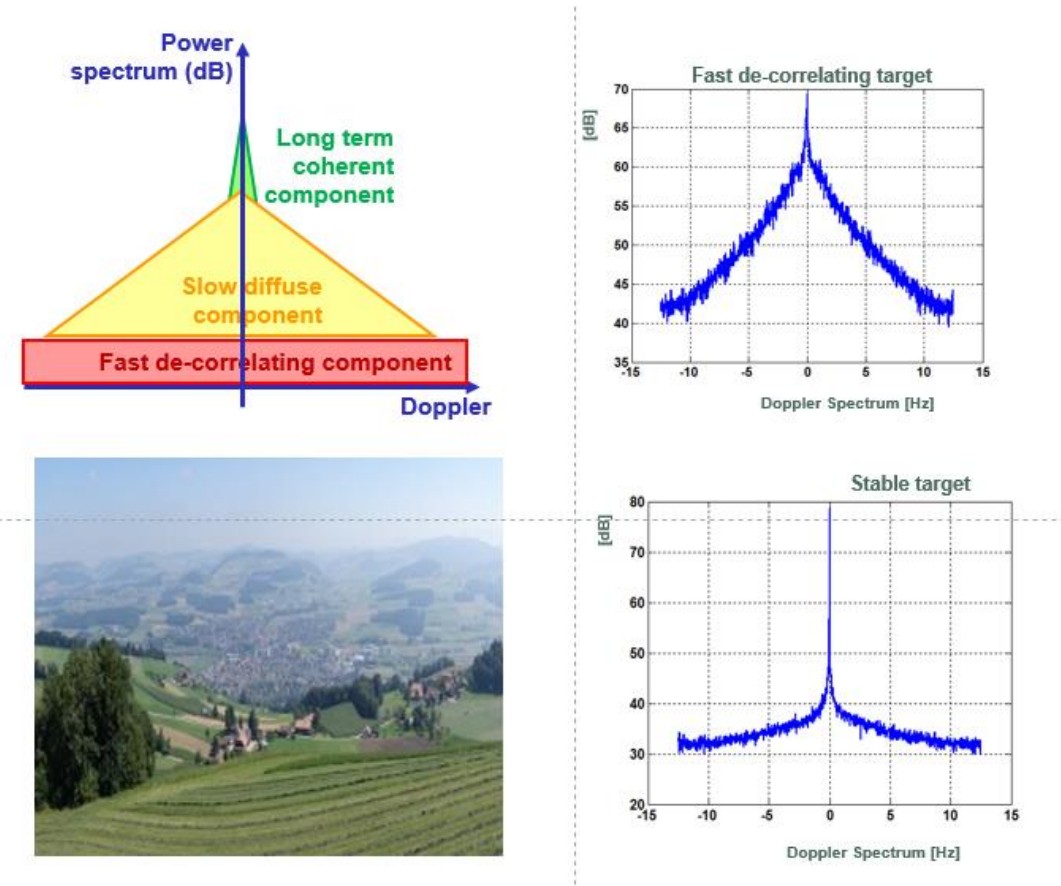

**Figure 1.** Intrinsic Clutter Model (ICM) model, top left, and measures of Doppler Power Spectrum Density (PSD) by Ku band Ground-Based Radar (GBR) in a rural scene (bottom left), of a fast decorrelating target, top right, and a stable target, (bottom right).

The ICM temporal coherence can be evaluated from the definition:

$$\gamma(\Delta t) = \frac{E[x(t)x^*(t + \Delta t)]}{\sqrt{E\left[|x(t)|^2|x^*(t + \Delta t)|\right]}} = \frac{E[x(t)x^*(t + \Delta t)]}{P_x} = r_x(\Delta t), \tag{4}$$

where $\Delta t$ is the time lag, $x$ the complex reflectivity of the decorrelating target, and $r_x$ the autocorrelation, which equates the coherence in virtue of the power normalization (3). The autocorrelation is the inverse Fourier Transform (FT) of the PSD (1):

$$r_x(\Delta t) = \frac{\lambda\beta}{4(\alpha + 1)}\left\{\int_0^\infty \exp\left(-\frac{\lambda\beta f}{2} + j2\pi ft\right)df + \int_{-\infty}^\infty \exp\left(\frac{\lambda\beta f}{2} + j2\pi ft\right)df\right\} + \frac{\alpha}{\alpha + 1}. \tag{5}$$

The temporal coherence is then obtained by combining (4) and (5):

$$\gamma_{\text{ICM}}(\Delta t) = \frac{1}{\alpha+1}\frac{1}{1+\left(\frac{4\pi\Delta t}{\lambda\beta}\right)^2} + \gamma_\infty,$$
$$\gamma_\infty = \frac{\alpha}{\alpha+1}. \tag{6}$$

For zero-time lag, $\gamma_{ICM}(0) = 1$, as it should be, while $\gamma_\infty$ is the constant or stable component.

Let us remark that the "stable" component is intended while the wind is blowing, whereas it has nothing to do with the long-term stability after the wind calms down.

To show this, we observe that the one-day coherence of a forest in C-band has been reported in the range of 0.2–0.4 by a wide and consistent literature [18–20]. This is by far not matching the values of $\gamma_\infty$ predicted by combining (2) and (6). The highest value of $\gamma_\infty$ corresponds to the lowest wind speed. We notice that for $\beta$ to be positive, and assuming the C-band, the wind speed is bounded from (2):

$$w \geq \frac{10^{-0.4147}}{2.2369} = 0.172 \text{ m/s}. \tag{7}$$

This is close to the lower bound $w > 0.25$ m/s mentioned in [16]. For such very calm condition, coherence from (6) is about $\gamma_\infty = 0.994$, that was never found so high in literature.

On the other hand, a coherence of 0.4 or lower should come out from (2) and (6) only for a wind continuously blowing at $w > 8$ m/s that is not at all realistic, just looking in the statistics in Figure 2.

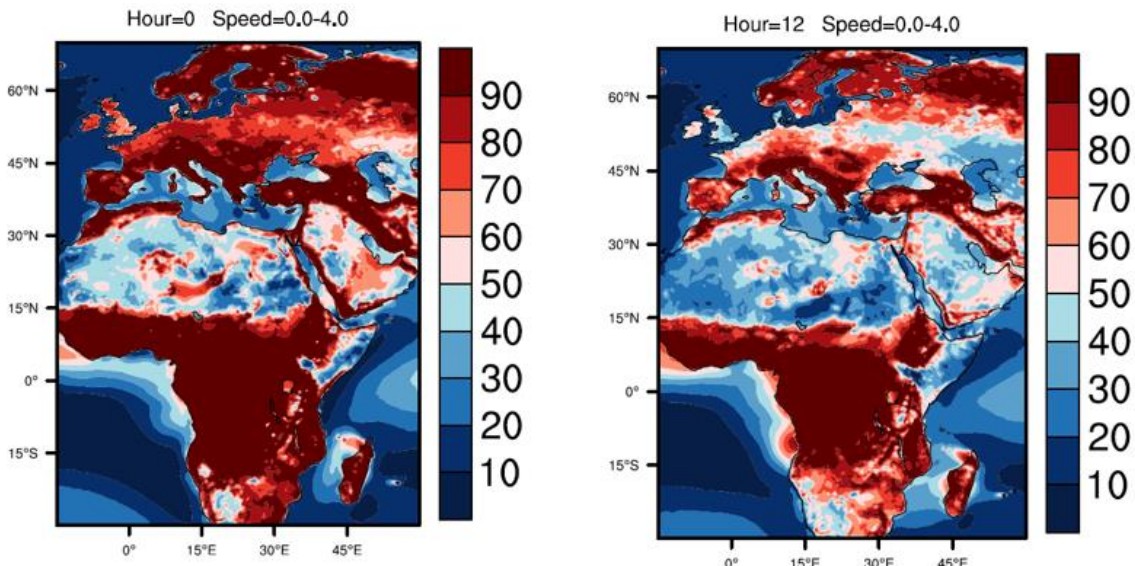

**Figure 2.** Probability of the surface wind speed $\leq 4$ m/s at hour 0 (**left**) and 12 (**right**), from ERA-5 (all data from 1989 to 2018). Courtesy of Yongjun Zheng, Jean-Christophe Calvet, METEO-FRANCE, CNRM (Centre National de Recherches Météorologiques). Notice that in forested areas wind speed over 4 m/s occurs for less than 10% of the time.

This is evidence that the ICM model is not suited for "long-term" (say one day) decorrelation.

### 2.2. The Random Walk Model

The exponential law is instead widely used for modeling temporal decorrelation in Radar echoes, over a wide variety of decorrelated targets [10–15]. In most cases, the exponential model is just assumed and validated. Indeed, exponential correlation results from the physics in random-walk, Markov, and telegraph processes [10].

Let us review the Wiener process, which models the decorrelation in many cases, and is suited to numerical implementation.

We assume that each of the many scatterers distributed in the resolution cell is subject to phase changes, from time to time, that may due to vegetation growth, vapor transpiration, morphological changes. As the target stays coherent for hours, or even days, we assume those changes to be so slow that we can discretize the time in steps. The contribution of the elementary scatterer at time $t_n$ to the complex reflectivity is:

$$s_p(t_n) = s_p(t_{n-1})\exp\left(j \cdot \Delta\varphi_p(n)\right), \tag{8}$$

where the $\Delta\varphi(n)$ is phase shift, which, eventually, can be related to a Line Of Sight (LOS) displacement $d_p(n)$:

$$\Delta\varphi_p(n) = \frac{4\pi}{\lambda}d_p(n). \tag{9}$$

If we assume that the scatterer amplitude does not change, then at the N-th time step, the scatterer contribution can be computed by the recursion (8), and the initial complex reflectivity, $s_p(0)$:

$$s_p(t_N) = s_p(0) \cdot \prod_{n=0}^{N-1} \exp\left(j \cdot \Delta\varphi_p(n)\right) = s_p(0) \cdot \exp\left(j \cdot \Delta\varphi_p(N)\right),$$
$$\Delta\varphi_p(N) = \sum_{n=0}^{N-1} \varphi_p(N). \tag{10}$$

The reflectivity in each resolution cell is the linear superposition of many elementary scatterers:

$$s(t_N) = \sum_{p=1}^{N_p} s_p(0) \cdot \exp\left(j \cdot \Delta\varphi_p(N)\right), \tag{11}$$

$N_P$ being the number of scatterers.

The temporal decorrelation is the result of the phase shift in (10):

$$\gamma_{RW}(N) = \frac{E\left[\sum_{p=1}^{N_p} s_p(0) \cdot \exp\left(j \cdot \Delta\varphi_p(N)\right) \cdot s_p(0)^*\right]}{\sqrt{E\left[\left|s_p(0)\right|^2\right]E\left[\sum_p\left|s_p(0)\right|^2\right]}} = E\left[\sum_{p=1}^{N_p} \exp\left(j \cdot \Delta\varphi_p(N)\right)\right]. \tag{12}$$

The correlation model (12) is quite general, the result would depend upon the assumptions on the statistics of the target change, $\Delta\varphi_p$. If we assume that phase or displacements variations are stationary, independent from step to step and Normal distributed, then $s(t_N)$ is a Random Walk (RW) in the class of the Wiener processes. The phase variance after N steps is the summation of N independent, identically distributed, random variables:

$$\sigma_\varphi^2 = \left(\frac{4\pi}{\lambda}\right)^2\sigma_d^2 \rightarrow \sigma_{\Delta\varphi}^2(N) = N \cdot \sigma_\varphi^2 = N \cdot \left(\frac{4\pi}{\lambda}\right)^2\sigma_d^2, \tag{13}$$

where $\sigma_d^2$ is the variance of the displacement in (9). The linear increase with N, or time, combined with the Normal distribution results in the exponential decorrelation:

$$\gamma_{RW}(N) = \exp\left(-\frac{\sigma_{\Delta\varphi}^2(N)}{2}\right) = \exp\left(-\frac{N}{2} \cdot \left(\frac{4\pi}{\lambda}\right)^2\sigma_d^2\right), \tag{14}$$

that can be converted into time, by assuming a time step $T_s$:

$$\gamma_{RW}(\Delta t) = \exp\left(-\frac{N}{2} \cdot \left(\frac{4\pi}{\lambda}\right)^2\sigma_d^2\right) = \exp\left(-\frac{|\Delta t|}{\tau}\right),$$
$$\tau = 2T_s\frac{1}{\sigma_d^2}\left(\frac{\lambda}{4\pi}\right)^2. \tag{15}$$

### 2.3. The Gaussian Model

The Gaussian decorrelation has been first proposed in [3] for SAR interferometry:

$$\gamma_G(\Delta t) = \exp\left(-\left(\frac{\Delta t}{\theta}\right)^2\right) \tag{16}$$

The corresponding PSD, is also Gaussian, for one known property of the Fourier Transform:

$$S_G(f) = \sqrt{\pi}\theta \cdot \exp\left(-\pi^2\theta^2 f^2\right). \tag{17}$$

The Gaussian power spectrum is indeed one of the first models for sea surface clutter, introduced empirically to fit observations in [1], and then widely adopted for ocean clutter [21,22].

### 2.4. The Generalized Random Walk Model

While the Gaussian model is suitable for scenes that are relatively fast-varying, the exponential decorrelation in (15) is the one mostly used for the long-term. A simple generalization is however needed to account for those elements in the resolution cell, that are stable over the very long-term, like branches or trunks, rocks, or the "Persistent Scatterers" [23]. The resulting decorrelation model has been widely used [10,13,14]:

$$\gamma_{gRW}(\Delta t) = \gamma_0 \exp\left(-\frac{|\Delta t|}{\tau}\right) + \gamma_\infty, \tag{18}$$

where the boundary condition $\gamma_{RW}(0) = 1$, should impose:

$$\gamma_0 = 1 - \gamma_\infty. \tag{19}$$

The corresponding PSD is derived by FT:

$$S_{gRW}(f) = \gamma_0 \frac{2\tau}{1 + (2\pi f \tau)^2} + \gamma_\infty \delta(f); \tag{20}$$

### 2.5. Fitting Decorrelation Models: Parameters Transformations

Let us summarize the models so far discussed:

1.　the ICM, defined by the PSD in (1) and the autocorrelation in (5);
2.　the generalized Random Walk (gRW), defined by the autocorrelation in (18) and the PSD in (20);
3.　the Gaussian one, whose autocorrelation (16) and PSD (17) can be updated to account for the long-term contribution, $\gamma_\infty$:

$$\begin{aligned}
\gamma_{RW2}(\Delta t) &= \gamma_0 \exp\left(-\left(\frac{\Delta t}{\theta}\right)^2\right) + \gamma_\infty, \\
S_{RW2}(f) &= \gamma_0 \sqrt{\pi}\theta \cdot \exp\left(-\pi^2\theta^2 f^2\right) + \gamma_\infty \delta(f).
\end{aligned} \tag{21}$$

We remark that the models have been derived in very different scenarios: the first for the wind-blown vegetation [17], the second for decays over days [10], and the third for the very short-term affecting ocean water [24]. Nonetheless, the models are sometimes used for different conditions than those mentioned in their original formulation, such as the Gaussian one, adopted for long-term SAR interferometry in [3,25], or the ICM used for a very wide class of targets, [26].

This fact suggests the possible interchangeable use of these models, which needs to find a correspondence between the two parameters. One parameter $\gamma_\infty$, is common to all, and defines the contribution of the asymptotically stable component. In the ICM, $\gamma_\infty$. derives from $\alpha$ according to (6).

The correspondence for the remaining parameter can be found:

by noticing that both the ICM and G coherences have a near parabolic behavior for $\Delta t = 0$:

$$\frac{1}{1 + \left(\frac{4\pi\Delta t}{\lambda\beta}\right)^2} \simeq 1 - \left(\frac{4\pi\Delta t}{\lambda\beta}\right)^2 \leftrightarrow \exp\left(-\left(\frac{\Delta t}{\theta}\right)^2\right) \simeq 1 - \left(\frac{\Delta t}{\theta}\right)^2, \tag{22}$$

that leads to:

$$\theta = \frac{\lambda\beta}{4\pi}; \ \beta = \frac{4\pi\theta}{\lambda}, \tag{23}$$

by equating the ICM and the gRW correlation after some significant decay, such as −1 Neper:

$$\gamma_{ICM} = \frac{1}{1 + \left(\frac{4\pi\Delta t}{\lambda\beta}\right)^2} = \exp(-1) = \gamma_{gRW} = \exp\left(-\frac{\Delta t}{\tau}\right), \tag{24}$$

that results in a very simple rule:

$$\tau = \frac{\lambda\beta}{4\pi}\sqrt{\exp(1) - 1} \simeq 0.1 \cdot \lambda\beta; \beta \simeq 10 \cdot \frac{\tau}{\lambda}. \tag{25}$$

This last fitting provides a physical meaning of $\beta$ as proportional to the ratio between the coherence time and the wavelength. Furthermore, as the exponential ICM autocorrelation has the same shape of the gRW PSD, then by imposing (25), we also ensure that the two power spectra will match at −1 Neper decay—that is a good addition.

An example of model fitting is shown in Figure 3 in the case of a target with a decorrelation time $\tau = 1$ s and a long-term stable coherence $\gamma_\infty = 0.2$.

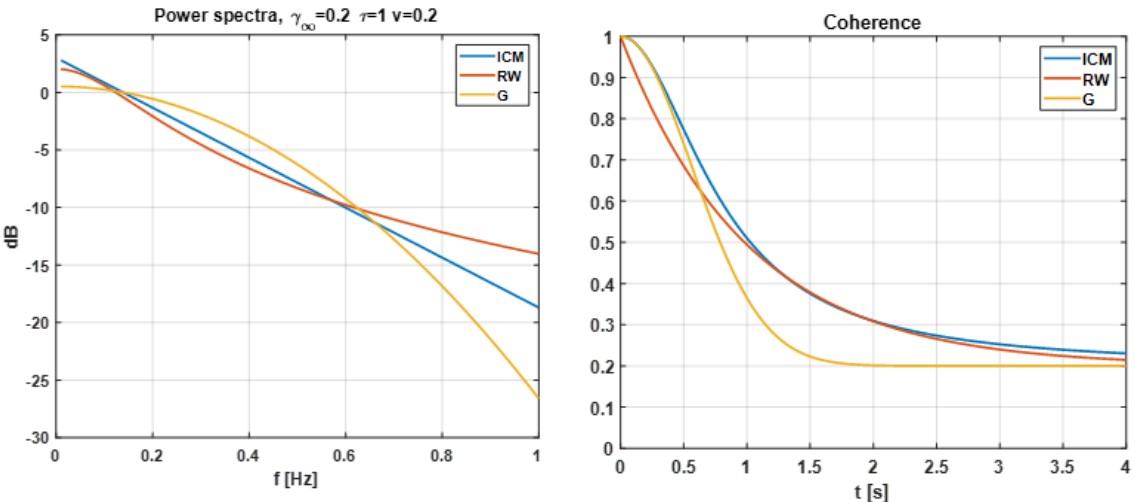

**Figure 3.** PSD (**left**) and autocorrelation (**right**) of the three models assumed: the Intrinsic Clutter Model (ICM), the generalized Random Walk (gRW), and the Gaussian one, plotted by imposing the fitting here proposed.

We notice that:

1.　the Gaussian model is loosely fitting with the other two, but for the ICM in the very short-term. This is a confirmation that the model is best suited for fast changes;
2.　the ICM model fits well the Gaussian in the short-term—as observed, but also the gRW in the long-term, confirming the goodness of the model transformation in (25).

One consequence of (25) is that we are now able to compute the decorrelation time implicit in the ICM model. We are here interested in the longest time constant that results by evaluating (25) for the largest $\beta$, which is found for the lowest wind speed (2):

$$\tau_{MAX} = 0.1 \cdot \lambda \cdot \beta_{MAX} = \frac{\lambda}{1.05 \cdot (\log_{10}(w_{min} \cdot 2.24) + 4.1)}. \tag{26}$$

For the wind speed limit $w_{min} = 0.2$ m/s, we get a time constant $\tau < 1$ s. Such decay time is orders of magnitude smaller than the value observed from classical Interferometric SAR (InSAR) literature [10,18–20].

### 2.6. Validation and Interpretation

Nowadays, there is consistent literature based on measures made by GBR, scatterometers, and spaceborne SAR to validate the models. They span a wide range of frequencies, from the Ku band, 17.2 GHz adopted by for most GBR [27], to the X, C, and L of past and present space-borne SAR and Unmanned Aerial Vehicle (UAV). All of these are precious sources of data and measures from the very short-term (for GBR) to the long-term (of months and years and on a global scale) for spaceborne SAR.

### 2.6.1. Ku Band Validation from GBR and SAR

At very high frequencies, the radar wavelength has limited or no penetration in the vegetation, and the reflection mostly comes from the lightest elements, grass, or leaves, which are quite influenced by wind. We note that we get a total decorrelation when the LOS phases of the elementary scatterers in the resolution cell are uniformly distributed within $\pm\pi$. This corresponds to a LOS displacement from (9) of $\pm\lambda/4$, which is ~0.75 cm in X-band and ±4 mm in Ku-band. This motivates the fast coherence decay, less than one day, measured from X-band Cosmo-SkyMed data [28].

An example of that is the 9-days multitemporal coherence map observed over a grass lawn in the mountain in June–July by a 17.2 GHz Ku band radar in Figure 4.

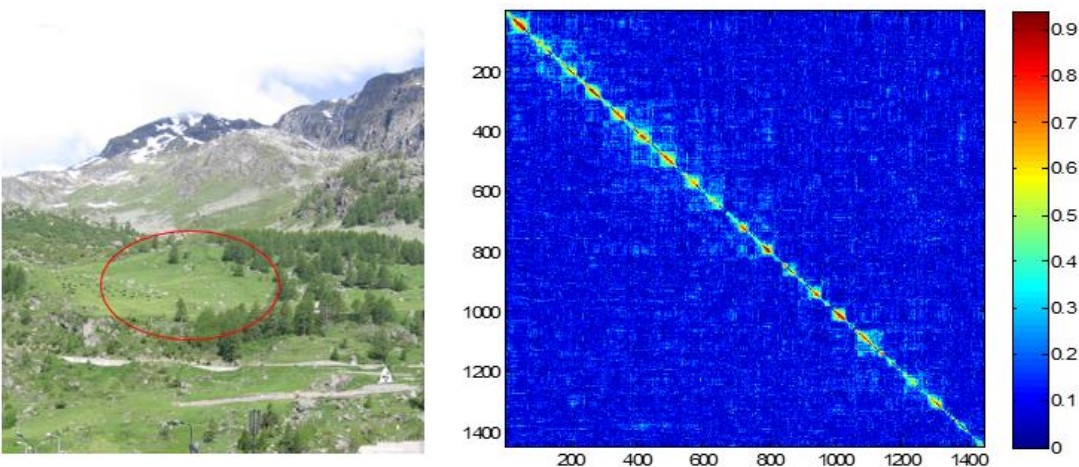

**Figure 4.** Multi-temporal coherence matrix, on the right, of the meadow encircled in red on the photo on the left, made by the position of the Ku-band (17 GHz) GBR. Numbers on the axis refers to the acquisitions, made every 20 min, starting from 20:05 of 18 June 2008. Coherence is found nighttime and not recovered from time to time.

It is also interesting to observe that in the intra-day, square blocks of highly coherent time intervals are found, almost systematically, on nighttime.

To better understand such behavior, Figure 5 shows four examples of coherence matrixes achieved over one day, in different scenes (forest/meadow), and different seasons. They have been taken as representative over a set of acquisitions campaigns within the project described here [29]. The scenes,

the instruments, and the processing were different from the one with the results in Figure 4. Nonetheless, one observes the same wide blocks of coherence at nighttime, which never occur in the daytime.

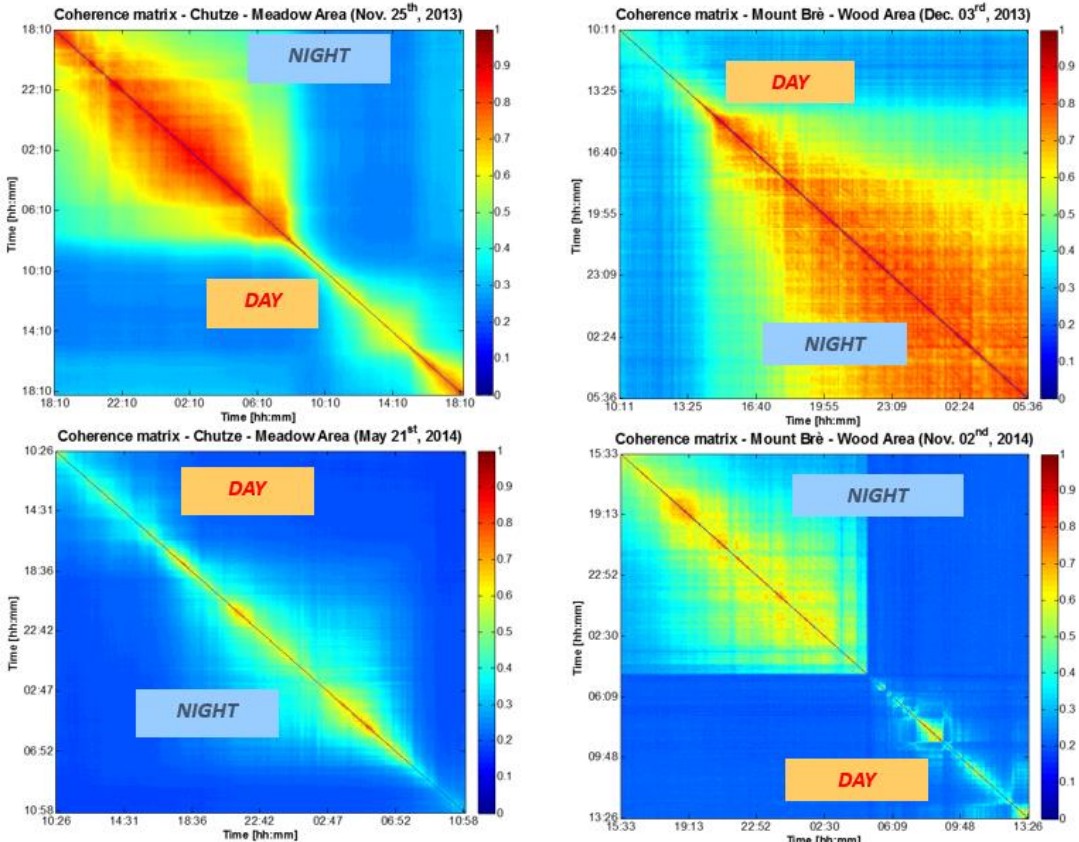

**Figure 5.** Example of coherence matrixes acquired over meadow (**left**), and forest (**right**) over about one day, in different periods of the year, by Ku-band GBR.

The evidence is that none of these coherence patterns can be attributed to the wind, as for the model predicted by the ICM, since, even with the slightest wind of 0.1 m/s, decorrelation time would be at most 5 ms, as from (26). Instead, we notice different decorrelation mechanisms, some of them are reported in Figure 6 that fit fairly well with the exponential gRW model and decay time of hours, which are orders of magnitude longer than the ICM decay. They may be attributed to effects triggered by sun cycle, and heat, such as sap-flows, vegetation growth, moisture, or water-vapor transpiration.

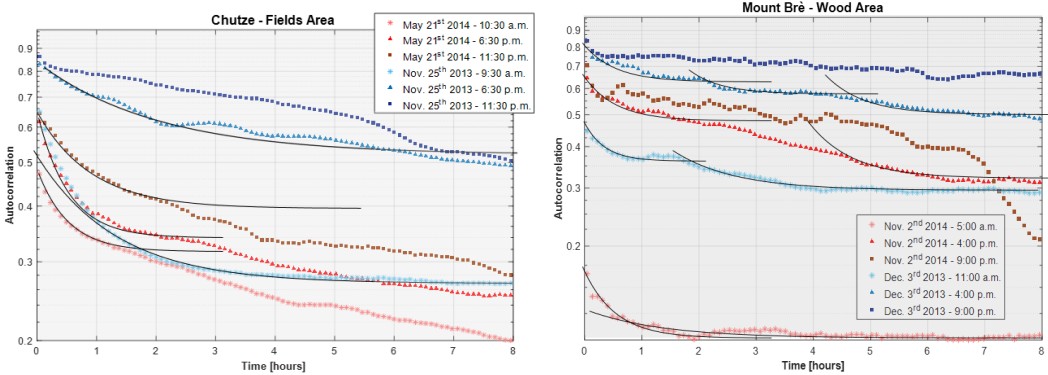

**Figure 6.** Time series of coherence over meadows (**left**) and forest (**right**) measured over eight hours for different times of the year, in the same two sites mentioned in Figure 5. The continuous line shows the fitting with exponential decays.

We observe that, also, the magnitude of the ICM decorrelation is quite different from the one shown in these examples. Even for a rare wind speed of 4 m/s (see Figure 2), we would get a considerable long-term coherence $\gamma_\infty > 0.34$, that is much better than the one in Figure 4 or Figure 5.

Notice that, more-or-less, any target is affected by a "fast" decorrelation, which occurs within the very first sample—that corresponds to 160 s in the acquisitions referred in Figures 5 and 6. These acquisitions were made by a real aperture GBR, described in [30], which was operated by interleaving wide-area scanning, in a mode named Scanning Aperture Mode (SAM), with the long sampling interval of 160 s, and a fixed pointing mode, named Fixed Clutter Mode (FCM), with fast sampling, 50 ms, but short duration. The two modes are described in Figure 7.

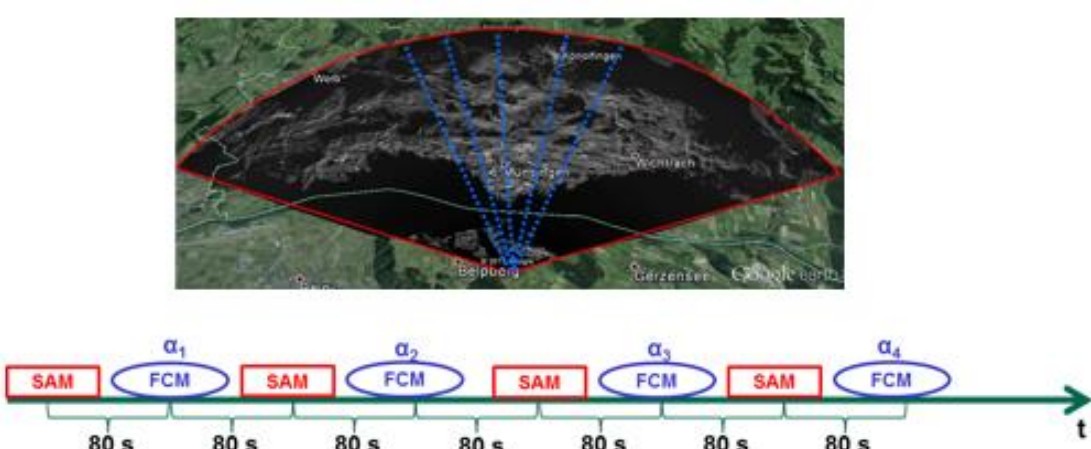

**Figure 7.** Operating modes of the real aperture, Ku-band radar used for the measures shown in Figures 5–9. Upper panel: example of a radar image. Lower panel: timeline, which results from interleaving every 80 s one observation at fixed pointing, named Fixed Clutter Mode (FCM), and one with an entire full aperture scan, named Scanning Aperture Mode (SAM).

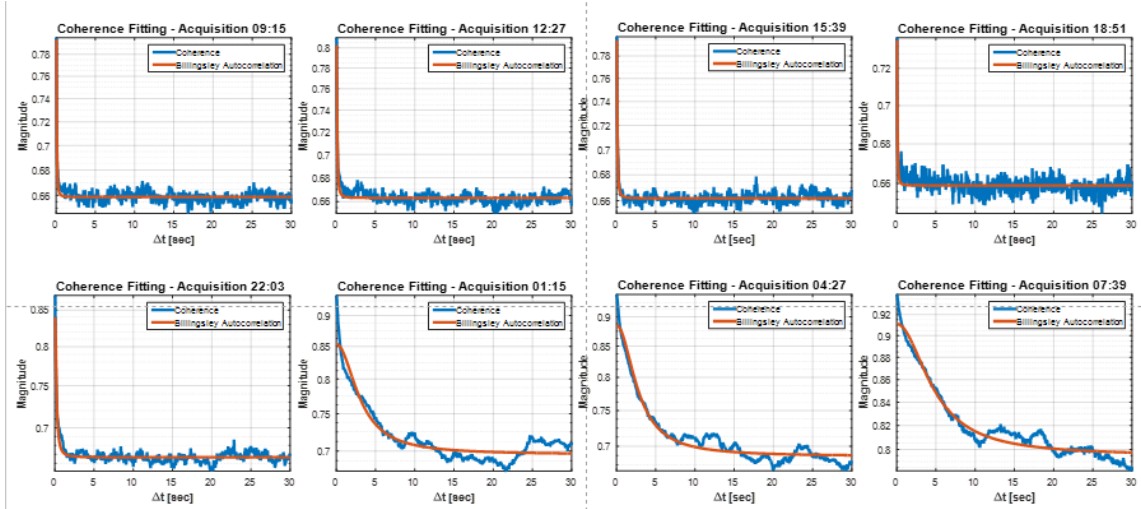

**Figure 8.** Coherence time series of a target over the forest, measured on 2 September, 2014, location Mount Bre (Lugano, CH), on the same target, in the short interval at different times of the day, superposed with the fitting ICM model.

Examples of the measures in the two modes, named SAM and FCM, are shown in Figures 8 and 9 for a campaign on a windy day (average of 10 m/s) in September, over a forested area near Lugano, Switzerland. The SAM observations in Figure 8 show signatures that totally decorrelate after just one

sample (=160 s), except for a couple of hours nighttime starting at 4:00. Instead, the FCM observations show a coherence step down to 0.5–0.6 (up to 0.85 at 4:00) that develops in much less than a second.

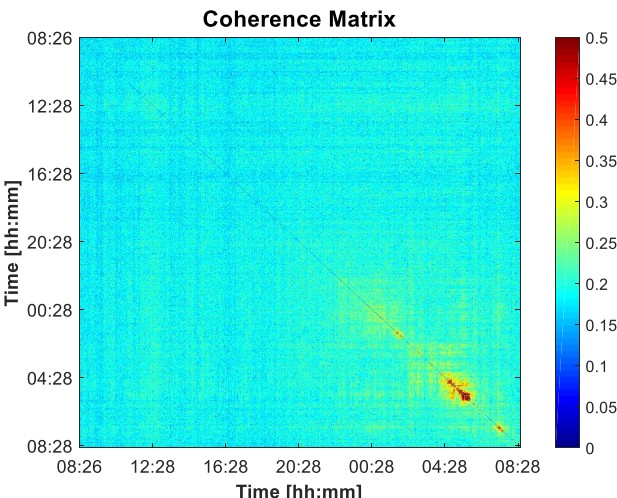

**Figure 9.** Long-term coherence matrix of the target considered in Figure 8.

Still, the most noticeable feature common in all the observations, shown in Figures 4, 5 and 9, is that coherence experiences fast drops at very precise time instants. These discontinuities can occur at the very first sample, like in the windy observations in Figure 9, or in other cases in Figure 5, not strictly related to the solar cycle (dawn, dusk). The randomness of the cut-off time instant is the evidence of a different mechanism, not developing according to the ICM or the gRW models discussed, but more probably a residual effect of a wind gust. This can be attributed to a spring-back of the vegetation after a wind-gust, repositioning closely, but not exactly.

Such residual displacements combine randomly in all the vegetation elements, resulting in a decorrelation (12). However, the residual displacements are the effect of the limited displacements allowed by vegetation, which cannot accumulate with the subsequent wind gusts, as for a "walk". It is rather a function of environmental conditions (season, temperature, humidity).

This motivates a decorrelation occurring at a random time step, but not increasing exponentially with time. Moreover, as a single wind gust is enough, we expect such a drop in any survey, but in flat calm, occurring mostly at nighttime.

This effect is not modeled by the ICM, whose role is to model correlation during wind, and not after.

### 2.6.2. P, L, C Band Validation

The long-term stability of vegetation increases with the longer wavelengths. This is due to the combination of two causes. The first is the reduced sensitivity to LOS displacements, the second is that the contribution to backscatter is much more due to the stable part of the vegetation, like branches, trunks, or terrain, then the light-weight, unstable, part, such as leaves or grass.

As for the short-term, the observation from a scatterometer over the boreal forest in [31] in L and P band seems to confirm the coherence loss in the presence of wind, compatible with the ICM model. It is rather clear that when wind ends, the ICM decorrelation is recovered to the original state.

The sudden drop of coherence, post wind gust, is somewhat implied in the C band analysis in [32], made by the same instrument and boreal forest scene of [31]. The authors observe the same two cases, high and persistent correlation in flat calm, and sudden coherence drop under the wind. Furthermore, the probability of the coherence drop was higher after 5 s than from 10 to 15 s. The fact that coherence drops suddenly at a random time, after a while, is an indirect confirmation of our wind-gust decorrelation.

The occurrence of that drop is implicitly modeled in literature by the zero-lag coherence, $\gamma_0$ in (18). The $\gamma_0$ value is implicitly estimated to be in the range of 0.2–0.4 for the 1 day revisit [18,20], and 0.2–0.5 for the 3 days revisit [19]. The step in case of short vegetation is less marked, as expected, as after 1-day a coherence 0.4–0.8 is well consistent with measures in [10,18–20]. The impression that a part of coherence drops in the short-term, the remaining is subject to slow decay is common in the community.

The long-term coherence decay is mostly an RW process with a very slow progression, due to many aspects, possibly liked to the vegetation cycle (growth, etc.), moisture, watering, farming, etc. The decorrelation time constant for the homogenous scene was found from days to a month [13], depending on the scene. An example of independent validation of the exponential decay is shown in Figure 10. Much more consistent validations are in [10,13], for C band, and in [11] for L, C, and X bands.

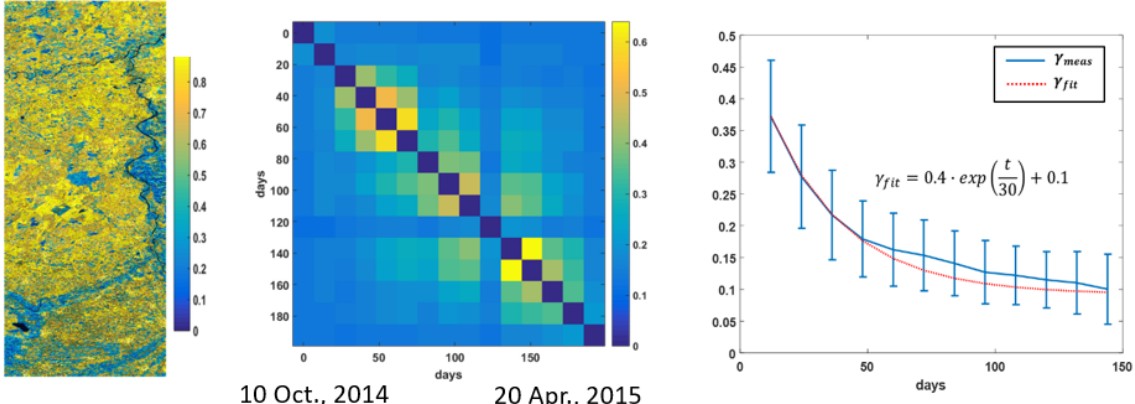

**Figure 10.** Long-term coherence of agricultural fields observed in C band, by Sentine1-1 SAR, every 12 days for about five months. (**Left**): an example of a 12-days revisit coherence. (**Mid**): matrix with the median of the coherences measured for each pair. (**Right**): fitting of the temporal decorrelation by the gRW model in (18).

In L band, a forest decorrelation close to 25 days, from [32], matches the qualitative comparison between the 1-day C band coherence, tandem European Remote Sensing (ERS) and the 44 days L band, Japan Earth Resources Satellite (JERS), over Borneo forests [33]. The quadratic scaling of decorrelation time with wavelength, predicted by the RW model (15), is another confirmation of the validity of the model, compared to the 1.2 power-law implicit in the ICM, from (2) and (25).

*2.7. Model Refinement: The Sum of Exponentials (SoE)*

So far, we analyzed and compared different decorrelation model. We have shown that the gRW model has some good properties, namely: to have simple and closed-form expressions for both coherence and PSD, to be able to fit reasonably other models, like the ICM, and to provide a very good fit for long-term decorrelation, as empirically verified. We underline that the exponential model has been widely validated and well adopted in literature [10–15].

The real word is different since many changes concur together to a vegetated target, including wind, moisture, sap-flow, vegetation growth, water-vapor transpiration, mixing of heterogenous targets in the volume, soil liquefaction, disasters [15,17,32,34–39].

However, if we wait enough (e.g., hours or a daily cycle), the most relevant effects are:

1.  a relatively fast drop of coherence, due to short time events, such as wind gust, that is not recovered anymore;
2.  a slow-varying temporal decorrelation that evolves in the long-term as a random walk;
3.  a long-term stable contribution.

The gRW model accounts for these three aspects, but to cope with the first, it needs to tune the initial decorrelation to values lower than the one predicted by (19) [10,11,13,15,20]:

$$\gamma_0 \leq 1 - \gamma_\infty, \tag{27}$$

that implicitly implies a drop in the coherence occurring in a very short time. While the short time behavior may be neglected, this should be necessary specified to compute the FT of the correlation and achieve a Power Spectrum that is positive or null.

A solution for this was proposed in [15], for pasture on drained peat soils. The model that we propose here is quite close to that, but accounts for the fast decorrelation by a short-term exponential:

$$\gamma_{\text{SoE}}(t) = \gamma_F e^{-\frac{t}{\tau_F}} + \gamma_0 e^{-\frac{t}{\tau}} + \gamma_\infty, \\ \gamma_F + \gamma_0 + \gamma_\infty = 1, \tag{28}$$

where $\gamma_F, \gamma_0$ models the fast and slow components, that decay with time constants $\tau_F$, $\tau$. This is very similar to the one proposed in [37], also with two exponentials, for ash detection over volcanoes.

The model is fully compatible with (18), which we retrieve for $t \gg \tau_F$:

$$\gamma_{SoE}(t) \simeq \gamma_0 e^{-\frac{t}{\tau}} + \gamma_\infty. \tag{29}$$

Moreover, it may well represent many scenarios that have been observed in real cases, by different scenes and frequencies, described in Section 2.5, while it keeps the advantage to have a simple closed-form FT, leading to the power spectra:

$$S_{SoE}(f) = \gamma_F \frac{2\tau_F}{1 + (2\pi f \tau_F)^2} + \gamma_0 \frac{2\tau}{1 + (2\pi f \tau)^2} + \gamma_\infty \delta(f). \tag{30}$$

An example of a coherence profile is depicted in Figure 11.

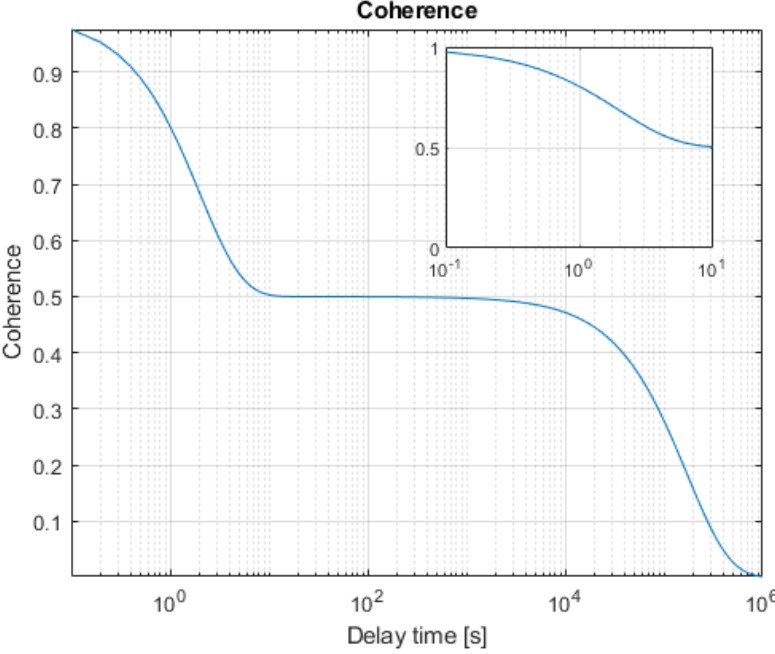

**Figure 11.** Example of an Sum of Exponentials (SoE) coherence profile as defined in (28) with $\gamma_F = 0.5$, $\gamma_0 = 0.5$, $\gamma_\infty = 0$ $\tau_F = 2\,s$ and $\tau = 2$ days. The top right box plots a zoom for the short-term.

## 3. Impact of Homogenous Clutter Decorrelation

The impact of a vibrating target to SAR focusing and interferometry can be evaluated formally in the case of a stationary, homogenous scene, similar to [40], but extended to the interferometric case and using the proposed SoE model.

### 3.1. Impact of Clutter on SAR Focusing

We assume, as in [29], the mono-dimensional SAR acquisition, along azimuth coordinate $x$. The raw data, $d(\xi)$, $\xi$ being the slow time, is the integration over the synthetic aperture, of the time-varying scene reflectivity:

$$d(\xi) = \int_{L_s} s(x,\xi) \cdot w(x,\xi) \cdot \exp\left(-j\frac{4\pi}{\lambda}R(x,\xi)\right)dx, \tag{31}$$

$(x, \xi)$ being the complex backscatter observed at azimuth x and slow time $\xi$, $w(x,\xi)$ the SAR antenna pattern on the scene, R the sensor-target distance and $L_s$ the synthetic aperture length. The scene dependence on $\xi$ is responsible for the clutter noise.

The focused data is the convolution of the raw data, $d(\xi)$ in (31) with the phase-matched filter, it is an estimate of the stable part of the complex scene, *s(x)*:

$$\hat{s}(x) = \int_{T_s} d(\xi) \cdot \exp\left(j\frac{4\pi}{\lambda}R(x,\xi)\right)d\xi. \tag{32}$$

The sensor target distance can be approximated by its first-order Taylor series:

$$R(\xi,x) = \sqrt{R_0 + (v\xi - x)^2} \approx R_0 + \frac{1}{2}\frac{(v\xi - x)^2}{R_0}, \tag{33}$$

where $R_0$ is the zero Doppler slant-range distance and $v$ is the speed of the platform. Then, by combining (31), (32), and (33) we get the estimate of the scene:

$$\hat{s}(x) = \int_{T_s} s(x,\xi) \cdot w(x,\xi) \cdot \exp\left(-j2\pi\frac{2vx}{\lambda R_0}\xi\right)d\xi = S(x; f_d) * W(f_d) \ \text{ for } \ f_d = \frac{2vx}{\lambda R_0} = \frac{x}{T_s \cdot \rho_{az}}, \tag{34}$$

Notice that $W(f_d(x))$ is the Impulse Response Function for the end-to-end SAR acquisition. If $w(x,\xi)$ is constant over the integration time Ts, its FT will be a sinc-like shape with lobe width $f_d \sim \frac{1}{T_s} \rightarrow \Delta x = \rho_{az}$ from (34). Therefore, if the scene is stable, its Doppler spectrum is impulsive $S(x; f_d) = s(x) \cdot \delta(f_d)$ then (34) becomes the reconstructed reflectivity blurred by the end-to-end Impulse Response Function (IRF):

$$\hat{s}(x) = s(x) * W(f_d(x)) . \tag{35}$$

Conversely, if the target at azimuth x vibrates with time, the case of interest, its contribution to the focused scene in (34), spreads over azimuth according to its FT. The defocusing is a random shape, which depends on FT of the moving target: $S(x; f_d)$. We can evaluate its variance, which is the expectation of the estimated backscatter:

$$\sigma_{\hat{s}}^2(x) = E[|\hat{s}(x)|^2] = W^2(f_d) * E[|S(x; f_d)|^2] = W^2(f_d) * S_d(x, f_d), \tag{36}$$

$S_d(x, f_d)$ being the PSD of the clutter at azimuth $x$. If we assume the clutter to be stationary all over azimuth, then

$$\sigma_{\hat{s}}^2(x) = W^2(f_d(x)) * S_d(f_d(x)). \tag{37}$$

Expression (37) shows that clutter noise starts to be noticeable, as its power spectrum gets broader than the system resolution.

In theory, the Signal to Clutter Ratio (SCR) should be the ratio between the power that falls within the resolution cell and the one that falls outside, due to the Doppler broadening. However, in the case of a stable scene, $S_d(f_d) = \sigma_s^2 \cdot \delta(s)$, one would get a limited SCR due to the sidelobes of the end-to-end SAR IRF, $W^2(f_d)$. Therefore, we define the SCR only basing on the Doppler spectrum:

$$SCR = \frac{P_s}{P_c + P_a} = \frac{P_s}{P_D - P_s + P_a} = \frac{1}{\frac{P_D}{P_s} - 1 + \frac{P_a}{P_s}} , \tag{38}$$

where $P_s$, $P_c$, $P_D$, and $P_a$ are respectively:

➢　the signal power, the contribution that falls within the azimuth resolution cell:

$$P_s = \int_{-\rho_{az}/2}^{+\rho_{az}/2} S_d(f_d(x))dx = \int_{-1/2T_s}^{+1/2T_s} S_d(f)\, df; \tag{39}$$

➢　the clutter power, that is the contribution that falls out of the resolution cell, up to the extent of the footprint:

$$D = \frac{\lambda R_0}{L} = \rho_{az}\frac{2vT_s}{L} = \rho_{az}B_aT_s, \tag{40}$$

　　$B_a$ being the Doppler bandwidth of the antenna;

➢　the total power in the footprint:

$$P_D = \int_{-D/2}^{+D/2} S_d(f_d(x))dx = \int_{-B_a/2}^{B_a/2} S_d(f)df, \tag{41}$$

　　where we assume that the footprint is limited by the antenna bandwidth;

➢　the alias power, that is the contribution that falls outside the footprint defined by the Pulse Repetition Frequency (PRF), that can be approximated as follows:

$$\begin{aligned}
P_a &= 2 \int_{D_f - \frac{D}{2}}^{D_f + \frac{D}{2}} S_d(f_d(x))dx, \\
D_F &= \frac{\lambda}{2v}R_0 \cdot PRF = \rho_{az}\frac{T_s}{2}PRF.
\end{aligned} \tag{42}$$

The SCR for a homogenous target is then evaluated by (38)–(42).

### 3.2. Impact of Clutter on SAR Interferometry

The second objective is to evaluate the performances related to interferometry. The interferometric quality is expressed by the modulus coherence between the master and the slave, that in this case, are two focused taken from the complex reflectivity of the same vibrating target in two different times. We model the two signals as rectangular windowing of the same process:

$$s_M(\xi) = s(\xi) \cdot \text{rect}\left(\frac{\xi}{T_s}\right), \tag{43}$$

$$s_S(\xi) = s(\xi - \Delta T) \cdot \text{rect}\left(\frac{\xi}{T_s}\right), \tag{44}$$

where $s_M, s_s$, being the master and slave images, generated from the same complex reflectivity $s(\xi)$ (we omitted the dependence on x for simplicity) and $\Delta T$ the interferometric revisit.

The coherence contribution due to clutter can be computed either in time or, for Parseval, in the frequency domain:

$$\gamma_{clut} = \frac{|E[s_M(\xi)s_S^*(\xi)]|}{\sqrt{\sigma_M^2 \cdot \sigma_S^2}} = \frac{|E[S_M(f_d)S_S^*(f_d)]|}{\sqrt{\sigma_M^2 \cdot \sigma_S^2}}. \tag{45}$$

The terms at the numerator can be evaluated as

$$
\begin{aligned}
E\left[S_M(f_d)S_S^*(f_d)\right] &= E[S(f_d) \cdot S^*(f_d)\exp(j2\pi f_d \Delta T)] * (T_s \cdot \sin c(fT_s)) \\
&= T_s \cdot (S_d(f)\exp(j2\pi f\Delta T)) * \sin c(fT_s).
\end{aligned}
\tag{46}
$$

The delay due to interferometric revisit, $\Delta T$, is a phasor in the frequency domain, which multiplies the Doppler spectrum of the clutter, $S_d$. The two terms combine in reducing the interferometric signal with the spreading of $S_d$ or the increase of the phase ramp, with the interferometric revisit. The rightmost term on (46), the convolution by the sinc pattern, accounts for the integration time due to focusing. That term is mostly irrelevant, as far as the integration time is small respect to the decorrelation time. This implies that the coherence does not depend on focusing time. We will show this in detail in the next section.

The normalization terms at the denominator in (45) can be evaluated as:

$$
\sigma_M^2 = \sigma_S^2 = E[S_M(f)S_M^*(f)] = S_d(f) * T_s \cdot \sin c(fT_s).
\tag{47}
$$

By substituting (46) and (47) into (45) we obtain the desired performance mode for interferometry:

$$
\gamma_c = \frac{S_d(f)\exp(j2\pi f\Delta T) * \sin c(fT_s)}{S_d(f) * \sin c(fT_s)},
\tag{48}
$$

where the term $\gamma_c$ stays for clutter coherence.

### 3.3. Performance Evaluation: Clutter and Coherence

We evaluate performance for focusing and interferometry, by primarily assuming the SoE model.

#### 3.3.1. Signal to Clutter Ratio

The signal is the power within the azimuth resolution, $P_s$, in (39):

$$
P_S = \int_{-1/2T_s}^{+1/2T_s} S_{SoE}(f)\,df = \frac{2\gamma_F}{\pi}\tan^{-1}\left(\frac{\pi\tau_F}{T_s}\right) + \frac{2\gamma_0}{\pi}\tan^{-1}\left(\frac{\pi\tau}{T_s}\right) + \gamma_\infty.
\tag{49}
$$

The power contribution $P_D$ is computed for (41):

$$
P_D = \int_{-B_a/2}^{B_a/2} S_{SoE}(f)df = \frac{2\gamma_F}{\pi}\tan^{-1}(\pi B_a\tau_F) + \frac{2\gamma_0}{\pi}\tan^{-1}(\pi B_a\tau) + \gamma_\infty.
\tag{50}
$$

The SCR derives from (38):

$$
SCR_{SoE} = \frac{1}{\frac{P_D}{P_s} - 1} = \frac{1}{\frac{2\gamma_F\tan^{-1}(\pi B_a\tau_F) + 2\gamma_0\tan^{-1}(\pi B_a\tau) + \pi\gamma_\infty}{2\gamma_F\tan^{-1}\left(\frac{\pi\tau_F}{T_s}\right) + 2\gamma_0\tan^{-1}\left(\frac{\pi\tau}{T_s}\right) + \pi\gamma_\infty} - 1},
\tag{51}
$$

where we have ignored the alias contribution.

In the case of the gRW model in (18), the *SCR* becomes:

$$
SCR_{gRW} = \frac{\tan^{-1}\left(\frac{\pi\tau}{T_s}\right) + \frac{\pi}{2}\frac{\gamma_\infty}{1-\gamma_\infty}}{\tan^{-1}(\pi B_a\tau) - \tan^{-1}\left(\frac{\pi\tau}{T_s}\right)}.
\tag{52}
$$

This expression can eventually be used to derive the *SCR* in case of ICM decorrelation. We assume the long integration time; therefore, $T_s \gg \tau$, as discussed in Section 2.1, and apply the parameter transformations in (6) and (25):

$$SCR_{ICM} \simeq \frac{\frac{\pi}{2}\alpha}{\tan^{-1}\left(\pi B_a \frac{\lambda\beta}{10}\right)} \approx 5\frac{\alpha}{B_a\lambda\beta} = 2.5\frac{\alpha L_a}{v\lambda\beta}. \tag{53}$$

This expression compares with the one in [40], for large *SCR* and in absence of alias, with the caveat that the *SCR* definition in (38) and (39) is slightly different from the one in [40].

In Figure 12, a map of SCR$_{SoE}$ has been represented by evaluating (51) for different combinations of fast decorrelation decay constants ($\tau_F$), the integration time ($T_s$), and the initial long-term coherence $\gamma_0$. Let us remind that $\tau_F$ is the decay time for the fast decorrelating clutter, which rules the delay, after which coherence drops to $\gamma_0$, and never recovers. This can occur within minutes. The worst condition in terms of SCR is when the integration time is longer than that, then $\tau_F \ll T_s$, and the coherence drop is significant, from Figure 12, $\gamma_0 \ll 0.6$. This could be the case of geosynchronous SAR in X, Ku, or higher frequencies, or C-band, but limited to forested areas.

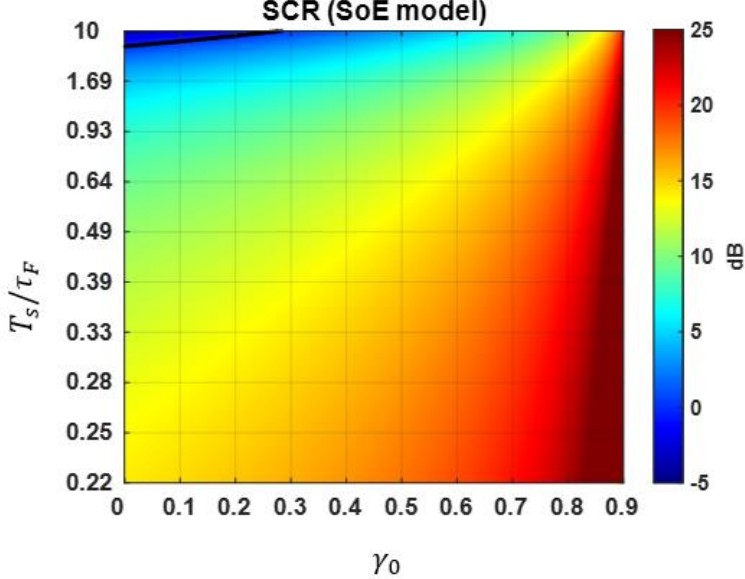

**Figure 12.** Signal-to-Clutter Ratio (SCR) achieved by focusing a target decorrelating according to SoE model, for different values of $\gamma_0$ (horizontal axis) and of the integration time normalized by the fast decay time (vertical axis). The long-term decay has been fixed to $\tau = 2$ days. The black line marks the combinations for which SCR is 0 dB.

The best condition, on the other end, occurs for SAR with short image time, such as Low Earth Orbit (LEO) SAR, or if the coherence drop is small like up to C-band SAR (for short vegetation), meaning that the clutter changed, but not so much. This is the case when both $\tau_F \gg T_s$ and $\gamma_0 \gg 0.6$.

### 3.3.2. Interferometric Coherence

The repeat-pass interferometric coherence can also be evaluated for different values of the parameters describing the decorrelation of the clutter, by substituting the SoE power spectrum, in (30), into the coherence expression in (48). The resulting coherence is represented in Figure 13 as a function of the ratios between fast decay constant and the integration time, $\tau_F/T_s$, and the one between the interferometric revisit and the long-term decay, $\Delta T/\tau$. The pure decorrelating target was assumed, $\gamma_\infty = 0$, and $\gamma_0 = 0.7$. One can observe that the final coherence is loosely related to the short-term time, or the synthetic aperture one.

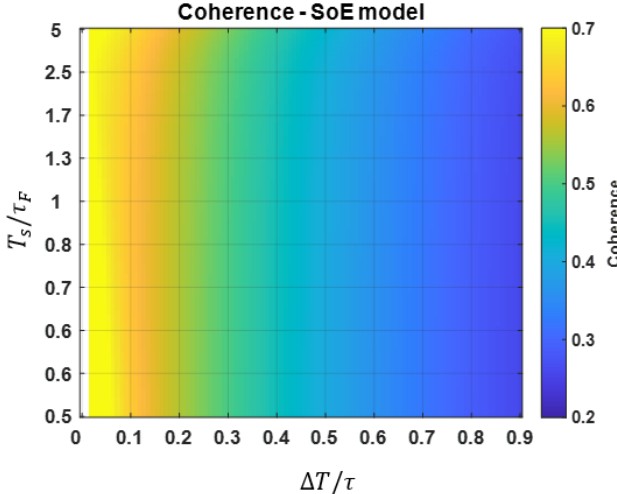

**Figure 13.** Expected coherence level as a function of the repeat pass interval, $\Delta T$, normalized by the long decay time (horizontal axis), and of the integration time, normalized by the shift decay time (vertical axis).

### 3.3.3. Comparison Between Short-Term and Long-Term Focusing

Here we discuss the cases of a short and a long integration time SAR, such as, for example, a LEO and GEO SAR. In the former case, the integration time is in the order of a second, and, in the latter case, up to an hour [41].

For the long integration SAR, $T_s \gg \tau_F$ and the SCR is the one in the upper part in Figure 12. The fast decorrelation is likely to occur during the acquisition; therefore, the focused image is affected by clutter, whose impact depends on the coherence drop, $1 - \gamma_0$. For short integration, the SCR is the one in the lower part in Figure 12. Focusing is achieved before that decorrelation occurs; therefore, the image is almost clutter-free.

As for interferometry, Figure 13 shows that the coherence is sensitive to the image revisit, $\Delta T$, but not to the focusing time, $T_s$. This means that coherence is independent on the focused image clutter. This is shown in Figure 14a, where both SCR and coherence are plotted versus $T_s$.

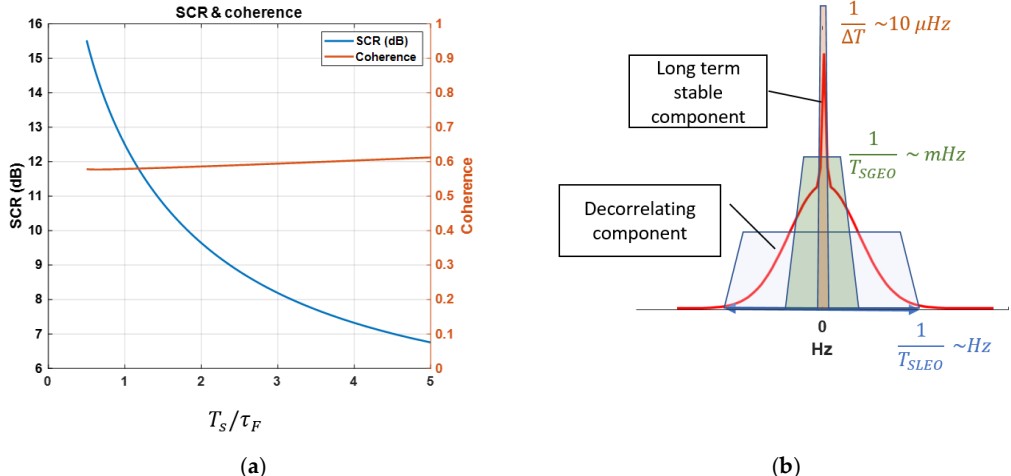

(**a**)  (**b**)

**Figure 14.** (**a**) Comparison between SCR and interferometric performances for the SoE decorrelation, as a function of the ratio between the focusing time and the fast decay time. Notice that coherence is mostly insensible to that. (**b**): Sketch of the scene power spectrum (red line), compared with the bandwidth of an LEO SAR (blue), a Geosynchronous Equatorial Orbit (GEO) SAR (green) and the one of the interferometric signal (orange), $1/\Delta t$.

An intuitive explanation comes from the analysis of the scene power spectrum sketched in Figure 14b. We remind that the clutter noise, from (38), increases with the power for Doppler frequencies $f_D > 1/T_s$, that in the order of Hz for a LEO SAR and mHz for a GEO SAR. Therefore, if $\tau_F$ is of a few seconds at least, no clutter is expected for the LEO SAR, but some for the GEO SAR. However, in both cases, the signal bandwidth is orders of magnitudes wider than the interferometric bandwidth, intended as $1/\Delta T$, 10 µHz for a minimum of 1 day. Clearly, the interferometric performance of the two systems will not be different, as shown in Figure 14a.

## 4. The Heterogeneous Target and Big Data Simulation

The increasing interest in new mission concepts with extended integration times, such as GEO SAR, requires that the effects of the scene decorrelation be carefully evaluated during the design phase to assess the expected impact on the data quality. The end-to-end simulation is a powerful technique exploited in SAR systems design, providing valuable information about the expected system performance, and facilitating the evaluation of trade-offs between different design solutions.

The end-to-end simulation can be either model-based or emulating the whole acquisition and processing chain. The first approach is suitable for early phases of new missions, to obtain a quick overview of the expected system performance. The second approach has a significantly higher computational cost, but provides very accurate results, fundamental during advanced phases of new missions for the fine-tuning of the different system parameters.

The clutter simulation approach discussed in the following sections is based on the Aresys Generic SAR Simulator (GSS) tool. The software provides Level-0 raw data according to the user-defined SAR system specifications. Each raw data line is generated by simulating the propagation of the transmitted Radar pulse from the satellite to each point target defined in the considered scene. The propagation simulation includes a gain term following the well-known radar equation plus a phase term proportional to the satellite-target distance. It is also possible to configure a set of additional effects such as propagation through the ionosphere or troposphere increasing the accuracy of the simulated data.

The expected simulation time for this approach is proportional to the number of echoes $N_{echo}$ and to the average simulation time per echo $T_{echo}$:

$$T_{sim} = N_{echo} \times T_{echo}. \tag{54}$$

The number of echoes is simply the product between the defined acquisition length and the system Pulse Repetition Interval ($N_{echo} = T_{acq} \times PRI$) while $T_{echo}$ depends on the defined scene (number of targets) on the effects to be simulated and on the system where the simulation is executed. The combination of long acquisitions, very high Pulse Repetition Frequencies, large or particularly dense (for very high-resolution systems) scenes can increase a lot the simulation time.

The GSS software is implemented according to a multi-thread, multi-process master-slave architecture allowing an improvement of the simulator throughput that, for big-data generation, is the major bottleneck. Figure 15 shows the simulation throughput as a function of the number of processes ($N_P$) and of threads per process ($N_T$) exploited. The reported results were obtained on a single system equipped with 4 processors Intel Xeon E5-4640 @ 2.40 GHz with 8 cores and 16 threads for a distributed scene of 28 million targets. The improvement is up to 25 times the original throughput if a single process single thread implementation is considered. The white region in the image represents a combination of $N_P \times N_T > 64$ where no tests have been performed due to machine hardware saturation.

The long integration times (up to hours) and the large imaged areas will further increase the simulation time for GEO SAR systems. For this reason, a Graphics Processing Unit (GPU) implementation of the GSS core, aimed at further improving the simulator throughput, is currently under development.

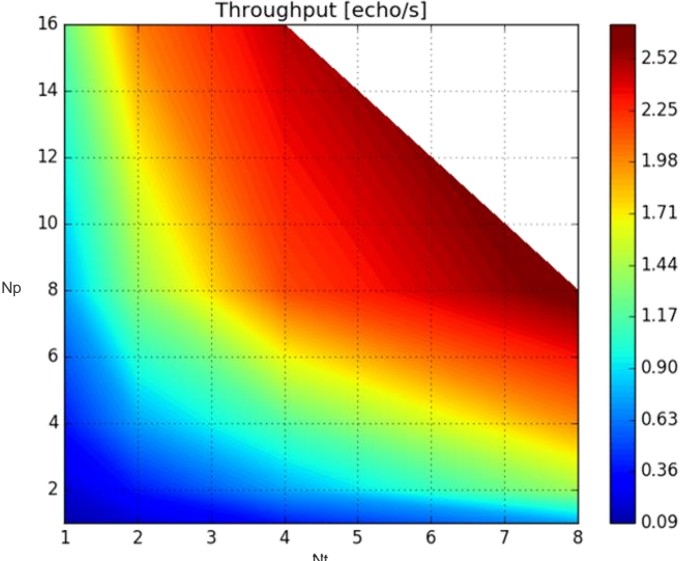

**Figure 15.** Throughput of the Aresys Generic SAR Simulator (GSS) tool as a function of the number of processes ($N_P$) and of threads per process ($N_T$).

### 4.1. Efficient Decorrelating Target Simulation

The simulation of the decorrelating raw is done by generalizing (31):

$$d(t, \xi) = \int s(\boldsymbol{P}; \xi) h_{SAR}(t, \xi; \boldsymbol{P}) d\boldsymbol{P}, \tag{55}$$

where the integral is extended to the area illuminated by the antenna main lobe, $\boldsymbol{P}$ is the location of each target within the scene, $t$ and $\xi$ are the fast and slow time respectively, $s(\boldsymbol{P}; \xi)$ is the temporally variant ground scene and $h_{SAR}(t, \xi; \boldsymbol{P})$ is the spatially variant SAR system Impulse Response Function.

The numerical implementation of (55) is carried out by simulating the decorrelating signature of each target, P. At the $n$-th time step, $\xi_n$, the target reflectivity can be computed by the linear combination of two circular Normal distributed samples, $w_1$ and $w_n$, with unitary covariance:

$$s(\xi_n) = \gamma(\xi_n) \cdot w_1 + \sqrt{1 - \gamma^2(\xi_n)} \cdot w_n, \tag{56}$$

where $\gamma(\xi_n)$ is the temporal decorrelation that we want to simulate, like those introduced in Section 2. The decorrelating source is then combined with the SAR acquisition model, in (55), which can include time and space varying ionospheric and atmospheric phase screen, amplitude patterns, etc.

An example of the simulation of more than 3000 decorrelating targets in C and X bands are reported in Figure 16, showing the targets average power spectra. The intended decorrelation was an ICM model, with a wind speed of 5 m/s and a PRF of 50 Hz, which was generated by implementing the gRW model in (18) with the parameter transformation in (25). The goal was to evaluate the suitability of the proposed approximation. The power spectra are plotted in Figure 16. The blue line represents the power spectra estimate by the simulated gRW process. The red and yellow lines represent respectively the gRW model (20) and the Billingsley ICM model (1), including five (bi-lateral) spectral replicas. The agreement between simulated spectra and models is good.

The black dashed lines represent the acquired scene bandwidth which can be exploited, as described in [40], to predict in advance the expected SCR in the focused data. The results of the simulation of the raw data according to (31) and of their focusing to get Single Look Complex (SLC) data are reported in Figure 17 for the targets in C (left) and X (right) band. The blue line represents the power level of the stable part of the focused targets. The red line represents the power level of the decorrelating part of

the targets. The predicted clutter level is represented with the yellow dashed line for the gRW model and with the purple dashed line for the ICM model. The clutter level predicted considering the two models is very close and in line with the results of the simulation. As expected, the SCR is worse in X band due to the lower value of $\gamma_\infty$ w.r.t. C band (0.42 vs. 0.60).

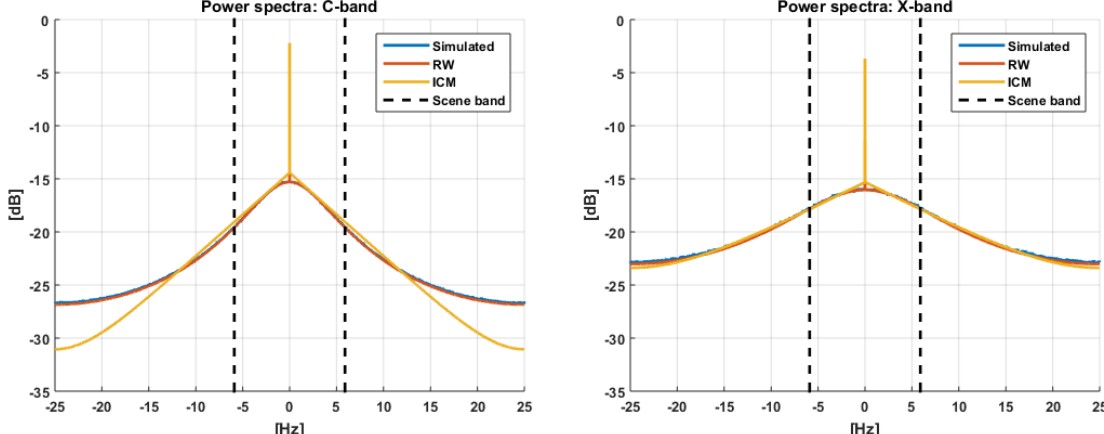

**Figure 16.** Power spectra of a set of decorrelating targets at C (**left**) and X (**right**) band. The blue line represents the average power spectrum of the targets. The red line represents the generalized Random Walk model spectrum. The yellow line represents the Billingsley ICM model. The black dashed lines represent the acquired scene bandwidth.

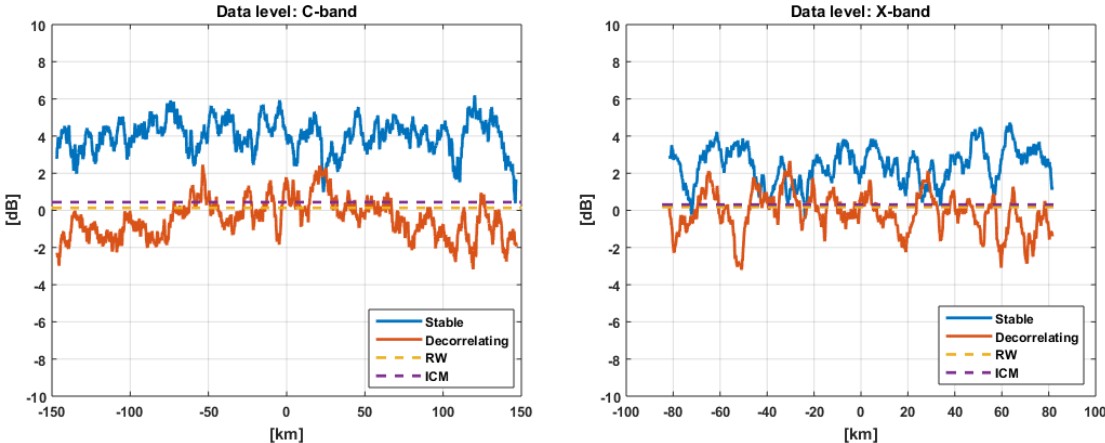

**Figure 17.** Simulated and predicted data levels for a set of decorrelating targets at C (**left**) and X (**right**) band. The blue line represents the power level of the stable part of the targets. The red line represents the power level of the decorrelating part of the targets. The yellow/purple dashed lines represent respectively the clutter power predicted from the gRW and the ICM model.

The scene decorrelation simulation approach is pictorially illustrated in Figure 18. The approach consists of four main steps:

1.  generate the Radar Cross Section (RCS) and of the weights to be used in (56), according to the selected model, the classification map, and the RCS map. Different decorrelation processes can be associated with different target types according to the classification map;
2.  compute SAR IRF: the SAR IRF is computed for each target and Pulse Repetition Interval (PRI) including the required gain, delay, and phase delay terms and according to the SAR system defined by the user (trajectory, acquisition mode, antenna patterns, atmospheric delays, etc.);

3.　evaluate target RCS: the RCS of each target at the current PRI is computed according to (56), properly scaled for the desired RCS;

4.　integrate target echoes: the echoes from all the targets at each PRI are summed together to get the final raw data matrix.

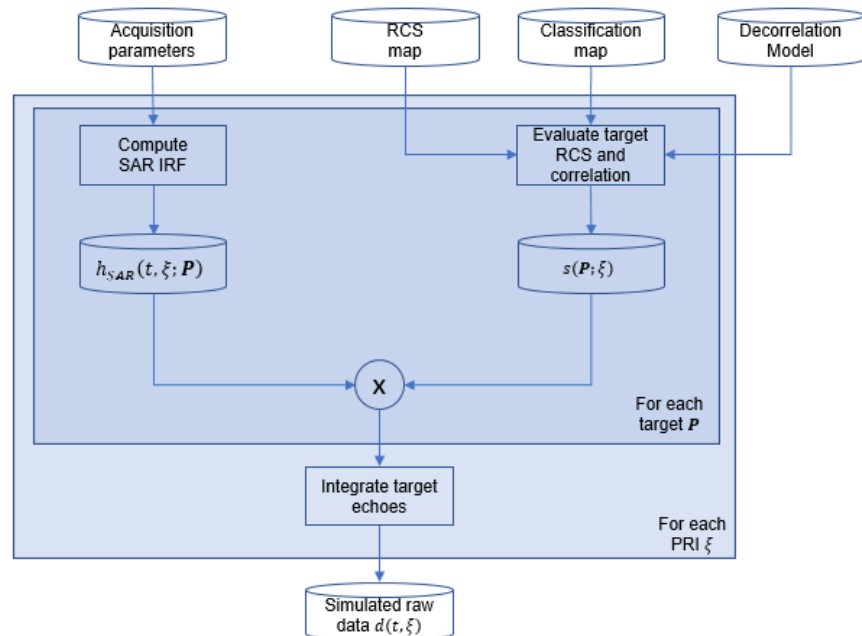

**Figure 18.** Flow chart of the scene decorrelation simulation approach.

*4.2. Impact of Targets Decorrelation on Long-Term Focusing: Exempla by Simulations*

The interest here is to appreciate the clutter noise introduced by the scene decorrelation during long-term focusing. In particular, we address two GEO SAR missions, operating at C and X-band. The systems' characteristics are reported in Table 1.

**Table 1.** GEO SAR system characteristics for scene decorrelation impact simulation.

| Parameter | C-Band | X-Band | Unit |
|---|---|---|---|
| Carrier | 5.405 | 9.6 | GHz |
| Azimuth Velocity | 23.2 | 23.2 | m/s |
| Range | 38,000 | 38,000 | km |
| Azimuth Resolution | 50 | 50 | m |
| Range Resolution | 20 | 20 | m |
| Integration Time | 900 | 450 | s |
| PRF | 50 | 50 | Hz |

The scene to be simulated has been derived by a Sentinel-1 Ground Range Detected (GRD) product acquired over North Italy on 1 February 2020. The quick-look (color composition of the co and cross-pol channels) of the GRD product is reported in Figure 19. The acquired scene has an extension of about 160 km in the azimuth direction (vertical direction of the QL) and 260 km in the range direction (horizontal direction of the QL). Note that, since GEO SAR LOS (main component South North) is orthogonal to LEO SAR LOS (main component East West) the azimuth and range directions will be swapped for the simulated GEO SAR data.

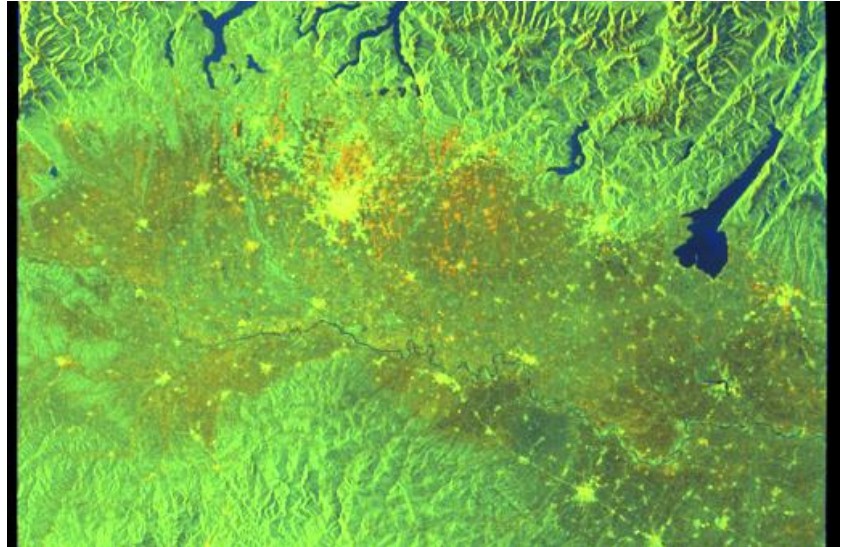

**Figure 19.** Quick-look of the Sentinel-1 GRD product exploited for the simulation of the decorrelating GEO SAR data scene.

The considered scene is quite heterogeneous including urban and rural areas, water bodies, and parts of the Alps. To account for the fact that the decorrelation process depends on the considered target, a classification map of the targets in the scene has been exploited. The map has been derived from the European Space Agency (ESA) Climate Change Initiative (CCI) land classification map providing the global land coverage classification with a 300 m resolution [32]. ESA CCI land classification includes 38 classes in total, here reduced to 5 for simplicity. The classification map of the five classes of targets to be simulated is reported in Figure 20.

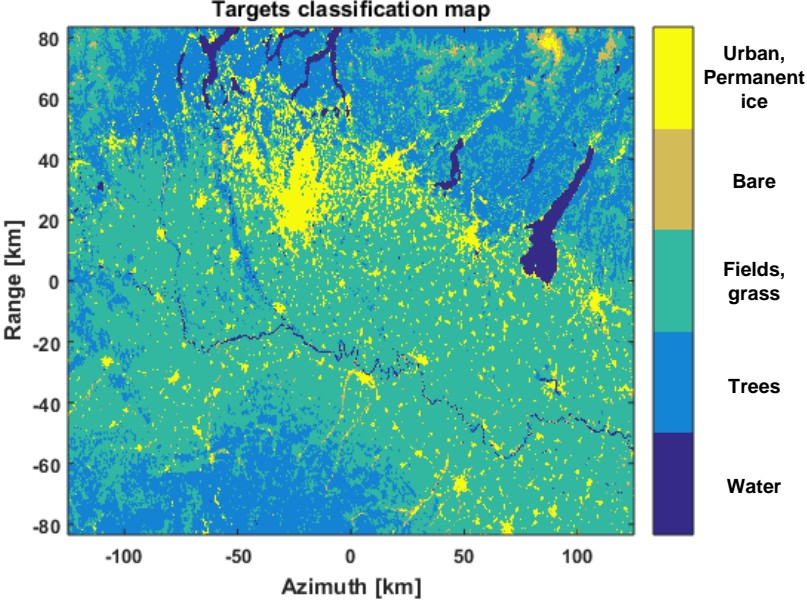

**Figure 20.** Classification map of the targets to be simulated.

We associated the classes with the parameters derived from the generalization of the ICM model in [26]. We used the gRW model since it is a very good approximation for the ICM, and even more appropriate for long-term focusing analysis. For the "Tree" class we assumed a wind speed of 5 m/s, which is a worst-case, according to Figure 2, and then derived the corresponding values of $\gamma_0, \tau, \gamma_\infty$ from the transformations in (6) and (26). The parameters resulting for the other classes are in Table 2.

**Table 2.** $\gamma_\infty$ parameters and performance for the GEO SAR simulation at C and X band.

| | C-Band | | | | | X-Band | | | | |
| | gRW Parameters | | | Performance | | gRW Parameters | | | Performance | |
| Class | $\gamma_0$ | $\tau$ [ms] | $\gamma_\infty$ | SCR [dB] | $\gamma$ | $\gamma_0$ | $\tau$ [ms] | $\gamma_\infty$ | SCR [dB] | $\gamma$ |
|---|---|---|---|---|---|---|---|---|---|---|
| Water | 0 | - | 0 | − | 0 | 0 | - | 0 | − | 0 |
| Trees | 0.4 | 36 | 0.6 | 16 | 0.6 | 0.57 | 20 | 0.43 | 16 | 0.43 |
| Fields, grass | 0.17 | 36 | 0.83 | 20 | 0.83 | 0.3 | 20 | 0.7 | 21 | 0.7 |
| Bare | 0.02 | 36 | 0.98 | 30 | 0.98 | 0.04 | 20 | 0.96 | 31 | 0.96 |
| Urban | - | 0 | 0.99 | $\infty$ | 0.99 | - | 0 | 0.99 | $\infty$ | 0.99 |

The result of the focusing of the simulated GEO SAR raw data is shown in Figures 21–23. In these simulations, thermal noise left at the same low value of Sentinel-1, to better highlight the effect of the clutter.

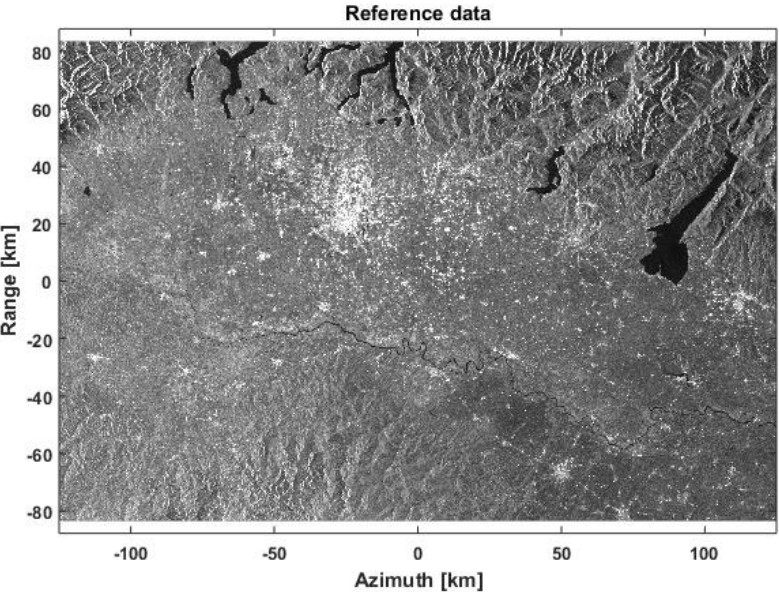

**Figure 21.** Simulated reference focused GEO SAR image. The target decorrelation effect is not included.

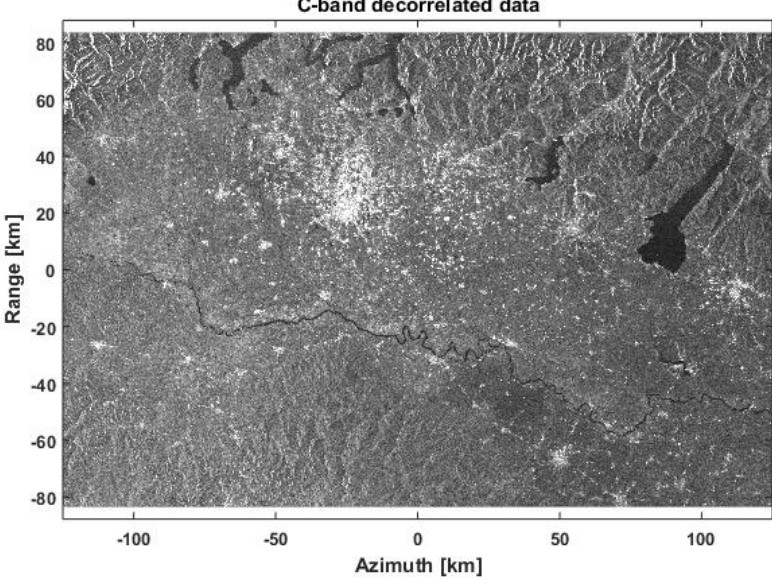

**Figure 22.** Simulated focused GEO SAR image in C-band with target decorrelation effect included.

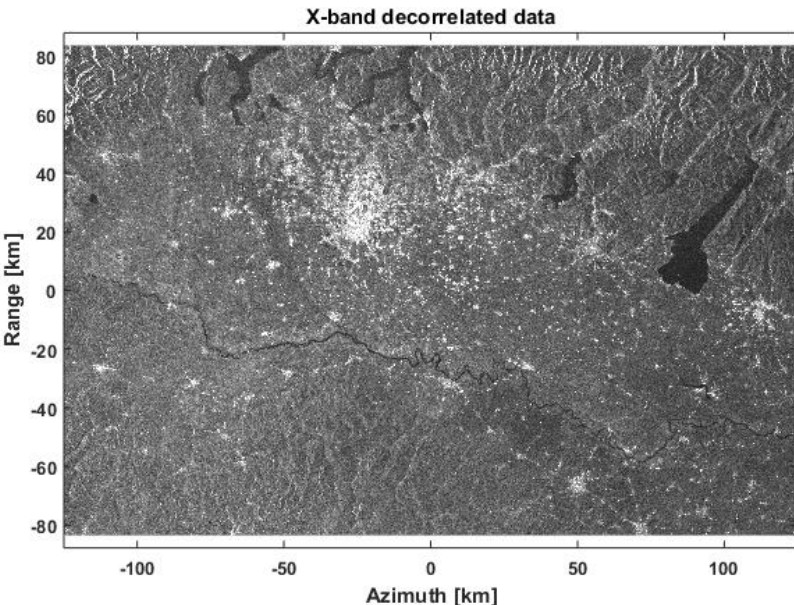

**Figure 23.** Simulated focused GEO SAR image in X-band with the target decorrelation effect included.

The image in Figure 21 represents the reference product obtained without considering any clutter. Figures 22 and 23 represent, respectively, the C-band and X-band GEO SAR focused image, where scene clutter was simulated according to Table 2. The difference between the two cases can be better assessed in the plot of Figure 24 where a transect of the simulated data is reported. The transect corresponds to about the km 10 in the range axis and spans the whole azimuth extension. The profiles obtained from the reference (blue), C-band (red), and X-band (yellow) data are quite similar except for the portion over the Garda lake (at about 90 km in azimuth). In that portion, the level of the reference data is around −22 dB, which corresponds to the Noise Equivalent Sigma Zero (NESZ) in the original Sentinel-1 image, while the level of the focused data at C-band is around −15 dB and slightly higher for the X-band data. This is consistent with performance predicted in Table 2, considering that the measure is due to the energy spreading from nearby targets. Also, note that the X-band profile is slightly lower in the other data portions.

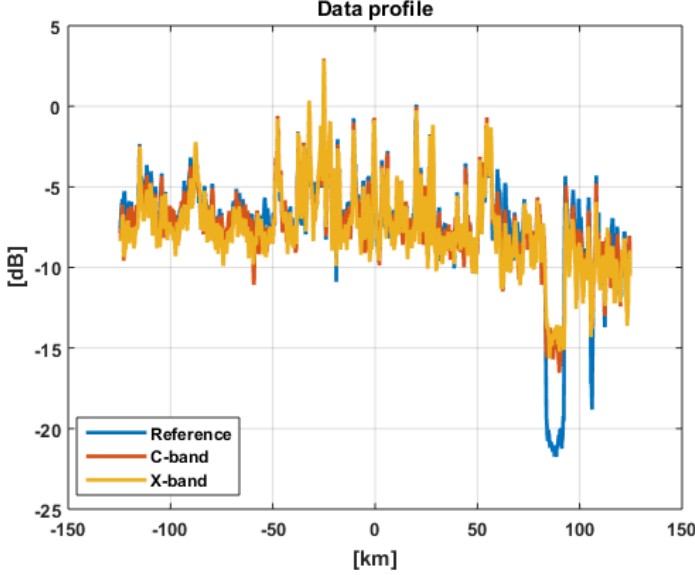

**Figure 24.** Comparison between the reference (**blue**) and decorrelated data profiles in C-band (**red**) and X-band (**yellow**) in the azimuth direction at about 10 km on the range axis.

Further, we have evaluated the interferometric coherence between the reference, in Figure 21, and the clutter affected images, in Figures 25 and 26, for C-band and X-band. As expected, the coherence is very low over water bodies and then changes according to the imaged scene content. In the simulation case, the coherence is high in the Po Valley (center of the scene) where most of the targets correspond to urban or field classes. The coherence then decreases over Alps (far range) and Apennines (near range) where woody areas are located. It is also possible to note that, as expected, the X-band coherence is lower than the C-band coherence, and in agreement with Table 2.

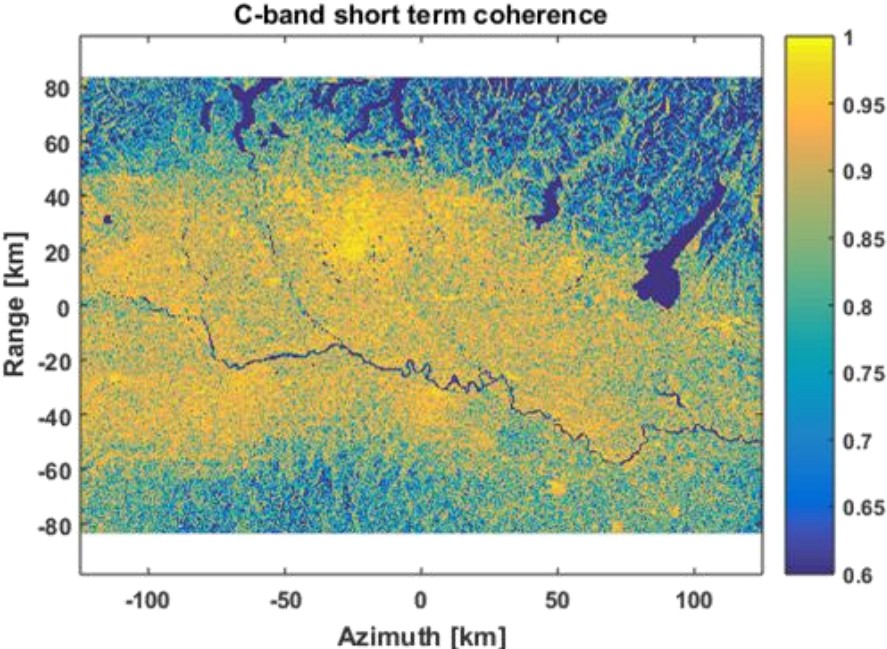

**Figure 25.** GEO SAR C-band simulation. Coherence between the reference image, shown in Figure 21, and the image affected by clutter, shown in Figure 22. The interferometric revisit is 900 s, see Table 1.

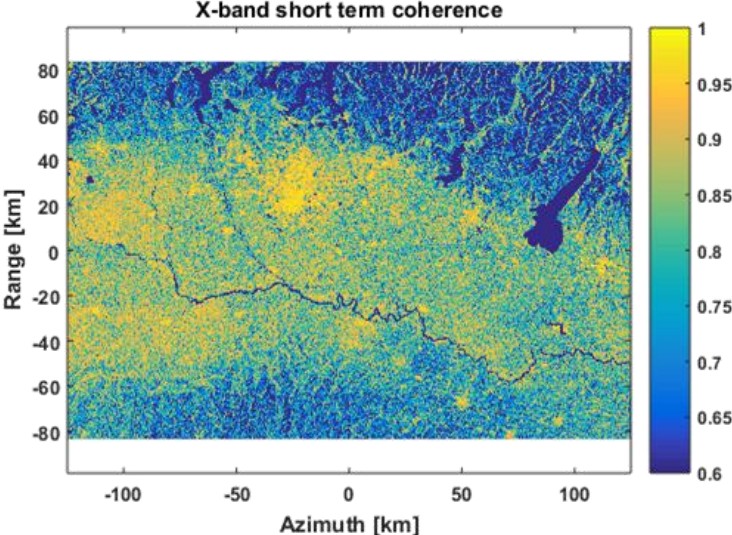

**Figure 26.** GEO SAR X-band simulation. Coherence between the reference image, shown in Figure 21, and the image affected by clutter, shown in Figure 23. The interferometric revisit is 450 s, see Table 1.

## 5. Discussion

The major innovation of the paper is the observation that—in a vegetated target, a step in the correlation is achieved in a very short-term; thereafter, a slow, long-term decay applies—as widely

known. This step has been attributed to the random displacement of the vegetation after a wind gust. The resulting time decorrelation profile is consistent with the following observations, made from literature and analysis of big data:

1.  the decorrelation develops in a fast interval and never recovers;
2.  there is only one step (the displacement is bounded by the elasticity of the vegetation);
3.  the decorrelation is not related to the sun cycle (dawn-sunset);
4.  decorrelation increases with the frequency, resulting in a total loss in Ku band ($\lambda/2 = 8$ mm);
5.  decorrelation is likely to occur in the short-term, up to a few minutes but not so for nighttime.

The Sum of Exponentials, another innovative contribution, accounts for that. Maybe it is not the best one to model the short-term evolution, but has the advantage to have a feasible power spectrum (semidefinite positive) and simple closed-form. Much more important, it is quite good for the long-term, the one we are interested in here.

The evaluation of interferometric performance is also innovative. An important conclusion is that, defined $\gamma_{th}$, $\gamma_s$ the coherence contribution of noise and scene (assuming to ignore other decorrelation sources), the total coherence would be:

$$\gamma = \gamma_{th} \times \gamma_s, \tag{57}$$

independently on the clutter noise that is generated by long-term focusing.

Another important conclusion is the observation that the ICM model is indeed very close to the gRW (exponential) one. However, the parameter transformation here proposed evidenced that the time decay, even for the lowest wind speed, is in the order of a second. Therefore, those decorrelations not specifically addressed by the model, that is forest while the wind is blowing, are more likely to come from a random walk. In that, the gRW Signal-to-Clutter-Ratio in (51) is more suited for grass, shrubs, short vegetation, or forest in calm conditions, than the one in [40]—that is to be used for wind-blown forests.

## 6. Conclusions

This paper reviewed the most used models for vegetated target decorrelation in mid-to-long-term SAR. Conclusions drawn from the analysis of big amount of data from P to Ku band, partly based on literature show evidence of at least two decorrelation mechanisms with time constants in the order of minutes and days.

A novel model was proposed, based on the Sum of two Exponentials decays. This model extends the other models, widely adopted in the SAR and InSAR community, while retaining the same simplicity and the formulation of temporal decorrelation and Doppler spectrum. The model has been used to evaluate both focused images' Signal-to-Clutter Ratio and interferometric coherence. It has been shown that the latter depends on the model parameters, but not at the former.

Finally, an efficient implementation of a SAR simulator for decorrelating distributed targets has been fully detailed. The simulation bases on the step-wide change in phase proper of Wiener Random Walks, at the base of the exponential decay. As such, it can cope with a very different scenario, such as heterogeneous scenes, and superposition of different mechanisms, such as atmospheric turbulence. The case of wind-blown clutter described by the well-known ICM decorrelation can be quite closely approximated by the parameters' transformations proposed in this paper. An efficient implementation is given, which is necessary to cope with huge data that a current LEO or a future GEO SAR could generate. Details and performance analysis are provided for this goal.

**Author Contributions:** Conceptualization, A.M.-G.; methodology and formal analysis A.R., D.G., M.M., S.T.; validation, A.M.-G., S.T.; writing A.M.-G., M.M., A.R. All authors have read and agreed to the published version of the manuscript.

**Funding:** This research was co-funded by Aresys srl, and by an activity performed in cooperation between Agenzia Spaziale Italiana, ASI, and Politecnico di Milano, in the frame of GEO SAR project (CUP F43C1700010005).

**Acknowledgments:** GBR data campaigns have been acquired by Aresys srl, and by Gamma Remote Sensing under ESA founded ESA-contract No. 4000108594/13/NL/CT GeoSTARe. We would like to thank Ing Antonio Leanza for data processing, and Fabio Rocca for his useful hints.

**Conflicts of Interest:** The authors declare no conflict of interest.

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
