# Peer review of "Vegetated Target Decorrelation in SAR and Interferometry: Models, Simulation, and Performance Evaluation"

_remotesensing, doi:10.3390/rs12162545_

Round 1

Reviewer 1 Report

The author proposed  a novel model that is an enhancement of the widely adopted exponential decorrelation, but accounts for the very fast changes occurring to vegetation blown by the wind, and the long-term persistent contribution.A formal method is developed to evaluate the performance of SAR focusing and interferometry on a homogenous, stationary scene, in terms of Signal-to-Clutter Ratio (SCR), and interferometric coherence.

The results look encouraging and motivating. But there are still some contents, which need be revised in order to meet the requirements of publish. A number of concerns listed as follows:

  • The abstract should be improved. Your point is your own work that should be further highlighted.
  • In the introduction, the authors should clearly indicate the contributions and innovations of this paper.
  • How is the complexity of the model? Please describe in detail.
  • The values of parameters could be a complicated problem itself, how the authors give the values of parameters in the model.
  • The inspiration of your work must be highlighted. For example, \  *10.1029/2012JB009178  *10.1109/TIM.2020.2983233  *10.3390/rs6064870  *10.1080/22797254.2017.1360155
  • The paper is in need of revision in terms of eliminating grammatical errors, and improving clarity and readability.

Author Response

The author proposed  a novel model that is an enhancement of the widely adopted exponential decorrelation, but accounts for the very fast changes occurring to vegetation blown by the wind, and the long-term persistent contribution.

A formal method is developed to evaluate the performance of SAR focusing and interferometry on a homogenous, stationary scene, in terms of Signal-to-Clutter Ratio (SCR), and interferometric coherence.

The results look encouraging and motivating. But there are still some contents, which need be revised in order to meet the requirements of publish. A number of concerns listed as follows:

  • The abstract should be improved. Your point is your own work that should be further highlighted.
  • In the introduction, the authors should clearly indicate the contributions and innovations of this paper.

We have partly rephrased the abstract, but, mostly important, we have added more details in the new “discussion” section.

Notice that the innovations are many: (1) the discussion about the sudden decorrelation and its just giustification, (2) the derivation of parameter transformation to fit the different models, (3) the proposed SoE model, (4) the analysis of the impact of clutter decorrelation (from focusing) on interferometry, (5) the criticism to the adoption of the ICM model outside the environment for what it has been derived.

For those reasons, we preferred to discuss all these points in the new “discussion” section, and, instead, keep the introduction compact.

  • How is the complexity of the model? Please describe in detail.

We have rearranged completely the section 2 in order to describe and better compare all the models, included the one proposed. The complexity comes from that.

  • The values of parameters could be a complicated problem itself, how the authors give the values of parameters in the 

We have also included this in the “discussion” section. Most of the parameters are taken from an analysis of data from iterature, there is indeed a wide and consistent set of analyses. We are confident to include the most important. Some of them have been added to the present version. As for the parameters used for simulation in section 4, the reference has been added.

  • The inspiration of your work must be highlighted. For example,\  *10.1029/2012JB009178  *10.1109/TIM.2020.2983233  *10.3390/rs6064870  *10.1080/22797254.2017.1360155

We have found the first and the third quite useful, and they have been added, the second paper has not a match with the topic, and the contribution of the last, useful for our goal, is already in the third one.

  • The paper is in need of revision in terms of eliminating grammatical errors, and improving clarity and readability.

We have cross-checked the paper many times, taking care to correct figure, table and equation numbering. Many typos were corrected, some thanks to reviewers. We are confident that the present version is very consistent.

Reviewer 2 Report

This paper presents a study on the decorrelation modeling in SAR images. The paper is well-written and the topic has actual meaning. The experiments are explicitly and the proposed SoE method seems effective. My only concern is that the methods and results are kind of mixed in section 2 and I think the authors should rearrange the section.

Author Response

This paper presents a study on the decorrelation modeling in SAR images. The paper is well-written and the topic has actual meaning. The experiments are explicitly and the proposed SoE method seems effective. My only concern is that the methods and results are kind of mixed in section 2 and I think the authors should rearrange the section.

We agree on that, in fact, section 2 has completely reformulated. We have now:

  • four subsections, one for each decorrelation model proposed,
  • one subsection for the comparison and parameter transformation rules,
  • one sub-section for the validation and critical discussion,
  • the last sub-section for our proposed model.

The text in each sub-section has been reworded. We think this structure is much more clear.

Reviewer 3 Report

Dear Authors
I started to read this draft with great interest as we all want to see comprehensive study about InSAR decorrelation and this draft seemingly has profound contents.It's difficult to follow the contents actually due to numerous mistakes in the draft with numbering , improper formatting, description of equations and figures.It's clear section four "simulation" parts has no cross check as all numbering is wrong.
In content wise, I would like to say few

1)  What model was used for the simulation in section 4? SoE?   I can't see an explicit statement. Seems that whole section 4 is somehow disconnected from other parts

2)  Figure 24 and 25, we know the coherence in urban areas should be far higher than other land covers. I can't see a distinguished difference in simulated coherence. Is the model working properly?Naturally I started to suspect. You have to identify why.

3) So the conclusion about the simulation result is correct? 

There are too many mistakes in the draft; thus I am worrying I caught only part of them.Please see my assignment and check the whole draft again.  I think current writing is closer to tech report rather than research paper.

 Regards

Author Response

Dear Authors
I started to read this draft with great interest as we all want to see comprehensive study about InSAR decorrelation and this draft seemingly has profound contents.It's difficult to follow the contents actually due to numerous mistakes in the draft with numbering , improper formatting, description of equations and figures.It's clear section four "simulation" parts has no cross check as all numbering is wrong.

We apologize for that. As answered to reviewer #1, we double-checked all the equation and figure numbering as well as the references. We corrected many of them, including all those evidenced by the reviewer. We thanks for that.

In content wise, I would like to say few

1)  What model was used for the simulation in section 4? SoE?   I can't see an explicit statement. Seems that whole section 4 is somehow disconnected from other parts

We acknowledge that section 4 was too much disconnected and report-style, in place of research. Indeed we made major changes. We largely abridged it by dropping inessential elements. We keep it now focused on the (1) efficient simulation of decorrelating target, (2) verification of the gRW and ICM model, (3) description of the implementation in a realistic environment, with heterogenous decorrelation and wide area, (4) example of clutter from geosynchronous SAR. We have added verification of results in table 2, see later.

Section 4.2 has been rephrased to better specify the model. We assumed the gRW (new acronym for generalized RW), discussing that choice, and we defined all the parameters in table 2.

2)  Figure 24 and 25, we know the coherence in urban areas should be far higher than other land covers. I can't see a distinguished difference in simulated coherence. Is the model working properly? Naturally I started to suspect. You have to identify why.

We understand that the text was not enough clear, and there were wrong figure numbering/referencing that did not help understanding. We have now made clear statement that those figures (now 25 and 26) report the coherence, between the focused image with clutter and the reference, clutter-free image. In particular, we remarked that the temporal baseline was not  is 450 and 900 s. Maybe the author is more familiar with the LEO revisit of several days, that is not at all the case here. Nonetheless, the differences are between urban, country, and vegetated areas are quite noticeable. To better represent them, we have reproduced the figure by saturating for γ<0.6.

3) So the conclusion about the simulation result is correct? 

Good suggestion. We rearranged Table 2, adding the expected performance, both in SCR and coherence. We think that there is a good agreement and we pointed out this in the text.

There are too many mistakes in the draft; thus I am worrying I caught only part of them.Please see my assignment and check the whole draft again.  I think current writing is closer to tech report rather than research paper.

We apologize for that, and corrected the paper. All the issues in the PDF were considered and were helpful. We discussed the innovation in the new “discussion” section. We agree that the paper reports some examples chosen from a wide set of acquisitions, and in this it seems a report. However, the is a proposed model, based on observations from natural behavior, then verification of results. It is quite important to evaluate performance. In fact, the way interferometric quality results, is somewhat unexpected for past literature, but explained. We strongly believe that observing the real behavior, formulate model, verify it and compare with the existence, is research.